

# An investigation of anthropogenic influences on hydrologic connectivity using stress tests

Amelie Herzog[1], Jost Hellwig[1], and Kerstin Stahl[1]

[1]University of Freiburg, Faculty of Environment and Natural Resources

**Correspondence:** Amelie Herzog (amelie.herzog@hydrology.uni-freiburg.de)

**Abstract.** Worldwide, human influences directly and indirectly threaten environmental flows through groundwater (GW) abstraction. This highlights the need to consider GW withdrawals together with climatic changes in future water management plans to maintain water availability in river networks. In alluvial geological contexts, such as the Dreisam River in South-Germany, contributions from GW often sustain streamflow in the summer months. In the specific case of the Dreisam however, several hydrological drought events between 2015 and 2022 lead to interruptions of longitudinal connectivity in the stream network. This raises the question on where and when the stream network is gaining or losing water from the GW and how these vertical connectivity changes influence streambed drying. This study therefore aims to analyse both, how changes in longitudinal and vertical connectivity in the Dreisam valley respond to stresses from climatic variations of recharge and to anthropogenic water abstractions from the hydrological system. As GW-SW interaction is difficult to measure, numerical groundwater modeling was used to obtain a spatial distribution of the exchange flow between GW and SW. The results show in which stream reaches, the connection between GW and SW ceases during dry conditions. Changes of vertical connectivity due to GW abstraction were found to be stronger than due to recharge stress on short timescales. A combined analysis of vertical and longitudinal connectivity depicts local points along the stream network, where the effect of GW abstraction on temporal drying dynamics is likely particularly strong. These results have to be interpreted within the limits of model reality and uncertainty. Simulated zero water levels were in good agreement with measured zero water levels at 50% of measurement locations, whereof the majority was in the valley bottom. Future work needs to improve coupling to upstream contributions and bedrock aquifers along the hillslopes of the valley. Overall, our findings highlight the value of a combined analysis of different dimensions of hydrologic connectivity for the evaluation of model results. Approaches that better distinguish locations affected by natural and anthropogenic drivers of hydrologic drought and streamflow intermittency deserve further development and are needed for application on different spatial and temporal scales.

## 1 Introduction

Expansion and contraction of non-perennial streams induce variations (in space and time) of hydrologic connectivity in three dimensions: longitudinal (upstream-downstream), vertical (surface-subsurface) and lateral (channel-floodplain) (Datry et al., 2017; Allen et al., 2020; Godsey and Kirchner, 2014; Freeman et al., 2007).The degree of hydrologic connectivity and the direction of exchange flows controls stream discharge magnitude, solute and contaminant transport, e.g. the dissolved organic





carbon (DOC) cycle and consequently also affects water quality and the lotic ecosystem processes (i.e. aquatic biota) (Datry et al., 2017; Costigan et al., 2016; Pringle, 2001). When flow ceases, hydrological connectivity is interrupted and physical, chemical and biological processes are modified. The disruption of hydrologic connectivity due to climate change or anthropogenic activities may induce long term changes of the river flow regime but also lead to cascading effects due to its influence

on fluxes of organic matter. As fluctuations of hydrologic connectivity are more severe in non-perennial compared to perennial river systems, the characterisation of spatial and temporal patterns of hydrologic connectivity is particular important for both, water resources management as well as ecosystem conservation purposes. At the same time, connectivity changes being the rule in non-perennial streams (and often non-linear), it is also more challenging to disentangle whether these changes are of natural origin or due to anthropogenic influences.

Among the major direct anthropogenic threats which cause hydrological alterations, such as lower flow volumes and prolongation of dry spells in rivers, are water withdrawals (Yildirim and Aksoy, 2022; AghaKouchak et al., 2021; Goodrich et al., 2018; Datry et al., 2017; Tijdeman et al., 2018). As water withdrawals locally create fluctuations of the water table, they modify vertical connectivity and thus, flow conditions in a stream are affected (gaining and loosing conditions). Particularly in snow dominated systems, vertical connectivity becomes more relevant in the summer season, as bank storage is less important

(Huntington and Niswonger, 2012). In order to understand if flow alterations in dry phases are exacerbated in response to water withdrawals, it is therefore crucial to investigate the relationship of longitudinal and vertical connectivity.

While many studies focus on one dimension of hydrologic connectivity, there are very few studies linking different dimensions of connectivity (Zimmer and McGlynn, 2018). Numerous studies on lateral connectivity exist, e.g. with a focus on hillslope connectivity (Zuecco et al., 2019; Blume and van Meerveld, 2015; M. Rinderer et al., 2019; Jencso et al., 2009) or connectivity

to floodplains (Xu et al., 2020; Czuba et al., 2019; Gallardo et al., 2014). Most recent studies on non-perennial river systems predominantly analyse longitudinal connectivity in order to describe flow characteristics (magnitude, frequency, duration) for clustering and characterisation of flow regimes or the spatio-temporal extent of non-perennial river systems (active drainage area,...) (Price et al., 2021; Hammond et al., 2021; Belemtougri et al., 2021; Botter et al., 2021; Botter and Durighetto, 2020). Statistical models have been used (Jensen et al., 2018) to describe flow dynamics in non-perennial stream reaches but those

models do not consider the physical processes underlying hydrologic connectivty.

Due to a lack of hydrometric data on non-perennial streams at necessary spatial resolution and over longer time periods, another prevalent research goal is the collection of streamflow and water level data at different scales, e.g. using field surveys (Zimmer and McGlynn, 2017), citizen science (Etter et al., 2020; Strobl et al., 2020) and new methods to measure dry phases in non-perennial streams (Herzog et al., 2022; Zanetti et al., 2021; Assendelft and van Meerveld, 2019; Jaeger and Olden, 2012).

But without the necessary data, linking the different dimensions of hydrologic connectivity remains difficult (Meerveld et al., 2020).

Specifically for an analysis of vertical connectivity, not only hydrometric data on the surface water (SW) system is required but also for the groundwater (GW) system. Joint approaches considering GW and SW data are lacking, partly due to low spatial data availability of GW heads but also because the measurement of transmission losses or GW-SW interaction itself is complex

and only possible at isolated points. Measurement approaches for the quantification of GW-SW interaction are often based on





tracers, e.g. heat (Angermann et al., 2012; Fleckenstein et al., 2010) or isotopes (Bertrand et al., 2014; Kalbus et al., 2006) which can be used to derive information on flow paths and residence times on very short timescales and small spatial scales only. Up to date, there is no reliable method for up-scaling of GW data to large scales (Foster and Maxwell, 2018; Barthel and Banzhaf, 2016). Therefore, data-driven studies on the spatio-temporal evolution of vertical connectivity of non-perennial

river systems hardly exist. Despite the scientific debate about the relevance of the GW term in the water balance (particularly in alluvial aquifers, where GW leakage is important) and the retroactive effects between headwater and lower catchment areas caused by GW leakage (Fan, 2019; Käser and Hunkeler, 2016; Covino and McGlynn, 2007), knowledge on the potential drivers of GW-SW interactions remains restricted to few locations.

By reason of the physically-based setup, integrated models (IM) are up-to-date a promising alternative for the representation

of GW-SW-dynamics at the catchment scale even though the possibility for calibration and validation of the simulated GW-SW interaction itself is limited as a result of missing data and required numerical effort (Barthel and Banzhaf, 2016). Besides limited possibilities of calibration and validation, uncertainties due to the influence of model resolution on model parameters and the complexity of the parameterisation are often discussed (Foster et al., 2020). While such parameter uncertainties are relevant when it comes to obtaining the best model results, they are less relevant if the focus is on process understanding. IM

have been successfully deployed to obtain a better understanding on control factors of GW-SW-interactions, for example by means of sensitivity analysis in order to investigate the influence of specific parameters on model results (Herzog et al., 2021a; Foster and Maxwell, 2018), hypothetic experiments to understand interaction of GW, SW and vegetation (Schilling et al., 2021) or stress testing, e.g. to assess the sensitivity of baseflow to antecedent conditions (Hellwig et al., 2021). Specifically stress testing or scenario-based modeling has traditionally been used for management purposes as the resilience of a system towards

specific stresses or even worst case scenarios is in focus and have recently been used in applied research in combination with sensitivity analysis. Most studies hereby inquire the effect of different climatic conditions (Hellwig et al., 2021; Stoelzle et al., 2014, 2020), while only few address anthropogenic influences despite their relevance for management purposes.

Here, we examine the sensitivity of (simulated) vertical connectivity in response to different indirect (recharge) and direct (groundwater abstraction) anthropogenic stresses and their combination at specific locations along a stream reach in a meso-

scale catchment in Southern Germany by means of model stress tests. We define metrics to describe the relationship of vertical and longitudinal (zero water levels, ZWL) connectivity following the hydrological alterations (HA) approach (Poff et al., 2010) and evaluate the stress test results using these metrics. The model experiments themselves target intraseasonal variability and how a change in vertical connectivity is related to ZWL occurences at specific locations in the catchment. A second objective is to analyse how this relationship may change in response to direct and indirect anthropogenic stresses added to the simulations.

The design of the model experiments and stress tests is chosen in order to answer specific research questions:

– What is the spatial distribution of longitudinal and vertical connectivity in the reference simulation, i.e. assumed near-natural conditions?

– Can we find metrics that describe patterns of longitudinal and vertical connectivity and their effects?





- How, in terms of these metrics, does vertical and longitudinal connectivity respond to GW withdrawals and recharge
stress?

## 2   Methods

### 2.1   Study area and data availability

The study area is the Dreisam valley $25\,\mathrm{km}^2$, a sub-catchment of the Dreisam catchment in the federal state of Baden-Württemberg in Southern Germany $577\,\mathrm{km}^2$ (Figure 1 a)). Gentle slopes in the center and increasing slopes and altitudes
towards the borders are characteristic for the Dreisam valley (Herzog et al., 2022). The geology is characterised by crystalline basement overlain by thick alluvial deposits. The uppermost alluvial materials belong to the so called Neuenburg formation, younger quarternary gravels with high hydraulic conducitivities ($3.2\,\mathrm{m\,d^{-1}}$ logarithmic). This alluvial filling reaches up to $25\,\mathrm{m}$–$40\,\mathrm{m}$ depth in the northern part whereas it decreases towards the southern part of the study area and contains the main aquifer (Wirsing and Luz, 2007). Older quarternary sediments with smaller grain sizes below this layer are less transmissive.
Previous studies on runoff generation processes suggest, that baseflow is one of the main runoff processes contributing to streamflow during dry phases in the Dreisam valley (Ott and Uhlenbrook, 2004). However, the degree of this connectivity varies and is difficult to quantify longitudinally.

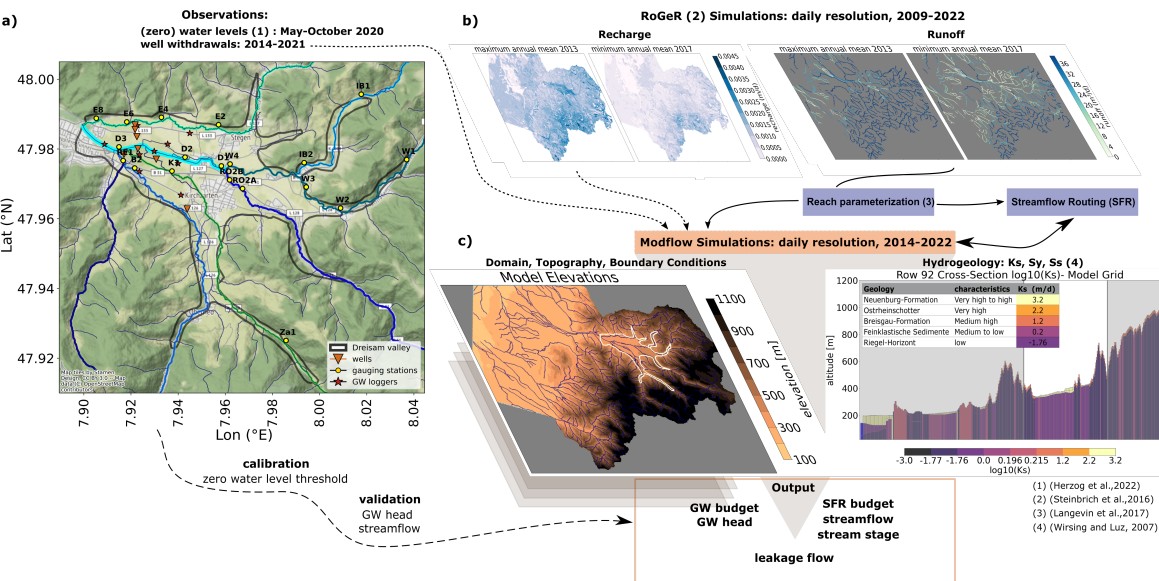

**Figure 1.** a) shows the Dreisam valley, the locations of the gauging stations from (Herzog et al., 2022) and the locations of the wells. b) RoGeR simulation output is converted to the model grid and added to the model as reach parameterization or boundary condition. c) The Modflow model uses data from a) and b) and additional hydrogeological data. Connections represented as dashed lines are input modified in the stress tests and bidirectional arrows indicate online coupling.





The aquifer is used for about two thirds of the water supply of the city of Freiburg im Breisgau ($8.6 \times 10^6 \, \mathrm{m}^3/\mathrm{yr}$). Based on data from government agencies on water withdrawals in the study area, it is estimated, that withdrawals for water supply account for $> 95\,\%$ of total withdrawals in the study area. Withdrawal data as well as GW level data are monitored and were provided by the largest regional water supplier (locations see Figure 1 a)). Other withdrawals were not available and were hence neglected in this model experiment. Stream stages and ZWL were measured at 20 locations in the study area (Herzog et al., 2022). The letters in the station-IDs refer to the different main tributaries they belong to, i.e. Dreisam, Eschbach, Brugga, Rotbach, Wagensteigbach, Ibenbach, Reichenbach Krummbach, Zastlerbach. Based on streamflow measurement campaigns in two years with contrasting climatic conditions 2020 (dry) and 2021 (wet), rating curves were developed to calculate streamflow at the specific locations.

## 2.2 The model concept

To represent both, the surface and the subsurface hydrological system we used a combination of the hydrological model RoGeR (Runoff Generation Research Model) (Steinbrich et al., 2021, 2016) and the GW model Modflow6 (Langevin et al., 2017) with surface water routing (SFR-package) (Figure 1). RoGeR is an advanced rainfall-runoff model, which calculates runoff components (interflow, overland runoff, percolation) for unit areas of similar climatic, topographic and pedological properties (Version RoGeR WB 1D). Modflow6 solves the three dimensional, transient GW flow equation (Darcy's law and continuity equation) for the simulation of GW heads. Surface water fluxes, well extractions and recharge are added in the form of boundary conditions. The model uses a (block-centered) control-volume-finite-differences method in order to give an (iterative) approximation of the analytical solution of the partial-differential equation. For unconfined conditions, the transmissivity of a grid cell varies based on saturated thickness of the cell.

The extension of the model domain covers not only the river valley but the entire Dreisam catchment and the neighboring catchment Möhlin-Neumagen ($708.09 \, \mathrm{km}^2$) with a river network length of $833 \, \mathrm{km}$ and the spatial resolution of the GW model is $100 \times 100 \, \mathrm{m}$ (Figure 1 c)). The model requires spatially distributed parameters determining surface and subsurface flows as well as timeseries of water input as an upper boundary condition. Time-variable and spatially distributed input to the Modflow6 GW model is taken from RoGeR simulation output and available in daily resolution for the time period $2009 - 2022$. The percolation component corresponds to recharge and the sum of interflow and overland runoff components corresponds to runoff directly contributing to streamflow. For the aquifer parametrization time invariant, gridded data of hydraulic conductivity and specific storage for the different layers of alluvial valley filling according to (Wirsing and Luz, 2007) are used. The empirical equation of Marotz allows to determine specific storage as a function of hydraulic conductivity (Fuchs et al., 2017; Marotz, 1968).

River reaches are defined as sections of streams in one model grid cell and further described by several parameters (such as reach width, depth, slope, thickness of streambed sediments, Ks of streambed sediments...). Details on parameterization requirements can be found in the Modflow6 documentation (Langevin et al., 2017). Parameters are derived from topography, river network data and governmental hydrographic surveying data.





In general, one model grid cell can contain several river reach units and thus, simulated GW head in one model grid cell is used to calculate the exchange flow (or leakage) of all the streams corresponding to this particular grid cell. The sign of leakage reflects the groundwater terminology and viewpoint of exfiltrating conditions (negative leakage) describing gaining streams and infiltrating conditions (positive leakage) describing losing streams. The leakage between the aquifer and the riverbed depends on the simulated stream stage and GW head for GW heads exceeding the streambed bottom. For GW heads lower than the streambed bottom leakage becomes independent of GW head and dependent on the water stage above streambed. Leakage is thus expressed as a product of hydraulic conductivity of streambed sediments, the streambed area of the reach with the difference between simulated stage and head (or streambed elevation) at the underlying node of the reach divided by the thickness of streambed sediments, which are all parameters of the stream network (Langevin et al., 2017).

Streamflow for each river reach is obtained based on the principle of continuity, considering that source terms (inflow from upstream reach, direct overland runoff and GW leakage to a reach) equal the sink terms (outflow to downstream reach, diversions from another reach, leakage to the aquifer). The leakage amount available per reach is determined by this stream reach water budget (assuming that there are no storage changes in the surface channels). Stream stage is the sum of stream depth and the streambed elevation. Calculation of stream depth at the midpoint of the stream bed is based on the calculation of flow at the midpoint. This flow is calculated first by means of the water budget and stream depth can be obtained subsequently using Manning's equation. As leakage flow is also dependent on stream stage, the equation is nonlinear and needs to be solved iteratively.

## 2.3 Evaluation of model results

In the following, the evaluation of the model results with respect to modeled leakage, ZWL days, longitudinal connectivity and the combined analysis of longitudinal and vertical connectivity is explained in detail.

Due to the lack of measurement data, a validation of modeled leakage is not possible. Instead, we verify that modeled and observed mean values of GW levels for the GW level loggers, streamflow (and stream stages) at the specific measurement locations agree. Negative outliers of stream stages may occur for GW heads far below the surface. For comparison, we therefore corrected extremely low minima and replaced them with the second minimum instead. Due to the availability of the ZWL dataset, it is possible to validate modeled ZWL days. Based on this validation, we pre-select a set of stations for which we can further analyse hydrologic connectivity. Hereby, we first determine a range of possible zero water level thresholds T for each station x within the sum of minimum water level and either 5 % (or 1 % depending on the variability) of the mean and the 90 % quantile (equation 1) of water levels h at this station.

$$T_x = [\min(h) + 0.05 * (\frac{1}{n}\sum_{i=i}^{n} h_i)...Q_{90}] \tag{1}$$

We then choose the threshold with the smallest difference in total number of observed and simulated ZWL days. If the so calibrated ZWL days and the general characteristics of the surface water system are comparable with the observations we assume, that the drying phases at these locations are well represented by the model.





**Table 1.** Relationship of ZWL days and leakage

| Information | ZWL days | leakage | Combination | Question |
|---|---|---|---|---|
| Duration | $t_{zwl}$: longest duration of ZWL days | $t_{0leak}$: longest duration of zero leakage days | $\delta t = \frac{(t_{0leak}-(t_{zwl})}{365}$ | How different are the duration of the longest period of ZWL longer and the duration of the longest period of zero leakage? |
| Timing | $t_0$: time of the first ZWL occurence | $t_{ch}$: time of the change to zero leakage prior to $t_0$ | $\Delta t = t_0 - t_{ch}$ | How long is the delay of the first ZWL after leakage ceases? Does zero leakage occur before zero water level? |
| Frequency | $N0_d$ and $N0_w$: mean duration of ZWL events for the recession (d) phase and for the rewetting (w) phase | $N0l_d$ and $N0l_w$ mean duration of zero leakage for the two phases | $\omega = \frac{(N0l_{d,w}-N0_{d,w})}{D_{d,w}}$  $D_{d,w}$: number of days in each phase | How long are ZWL and no leakage periods during during the recession and rewetting phase? |

To evaluate longitudinal hydrologic connectivity in the following, we use the metrics developed to describe the hydrologic regime of non-perennial rivers (Magand et al., 2020; Costigan et al., 2016; Gallart et al., 2012). From an ecohydrological perspective, three (hydrological) aquatic phases (dry, standing, flowing) relate to five (ecological) aquatic states (Meerveld et al., 2020; Datry et al., 2017; Gallart et al., 2017, 2012). The standing water phase is characterised by stagnating water and the formation of isolated pools. The water level dataset of (Herzog et al., 2022) contains data with a focus on the determination

of ZWL occurences and thus, the dry phase only. The raw dataset of ZWL occurences ($15\,\mathrm{min}$ resolution) was converted to ZWL days respecting a maximum duration ($<24\,\mathrm{h}$) between two ZWL occurences.

The direction (and quantity) of vertical connectivity of a stream reach can change with time depending on changing flow conditions in the stream and on changing GW head.

If GW leakage changes affect the drying of the streambed we assume, that the connection between GW and the streambed is

lost at first, i.e. a GW leakage change to zero is observed. Thus, we particularly focus on direction changes to zero leakage. Due to the reliance of simulated leakage on simulated stream stage, we can not evaluate the relationship of the two simulated variables. Instead, we compare simulated zero leakage and the measured ZWL days. As for ZWL, the definition of zero leakage also requires the use of a location-specific threshold $T_{leak}$. We define $T_{leak}$ as  10% of the maximum leakage simulated at each location.

In order to analyse the inter-dependency between vertical and longitudinal connectivity, we develop metrics for the combined analysis of the relationship between longitudinal and vertical flows based on the measured drying patterns and the simulated GW leakage direction changes. We analyse duration, timing and frequency of the different phases of vertical and longitudinal connectivity (Table 1).





## 2.4 Stresstest scenario definition

We defined different stresses due to the impact of climatic conditions through their different recharge conditions in the catchment (Figure 2 c) and e)) and stresses due to the anthropogenic impact in the form of GW withdrawals (Figure 2 b),d) and f))in the valley bottom near the main river. GW recharge is the quantity of water percolating from the surface-water system into the GW system and is directly linked to both, hydro-meteorological conditions (precipitation and evapotranspiration) and soil properties (infiltration capacity and runoff pathways). We ask the question of how sensitive GW-SW interactions are to

(i) changes in groundwater recharge and (ii) changes in GW abstractions from wells in the valley and (iii) how these stresses together exacerbate changes in GW-SW-interaction. Stress tests of GW water withdrawals are data-based ($2014 - 2022$). Synthetic, gridded recharge stress is generated from the simulated recharge output of the RoGeR model ($2009 - 2022$) in order to investigate the impact of different recharge magnitude (Figure 2 (c)) and seasonality (Figure 2 (e)). The magnitude scenario focuses on the impact of drought years occuring at a high frequency (each second year). The seasonality scenario expresses a

system change with a tendency for winters to be drier.

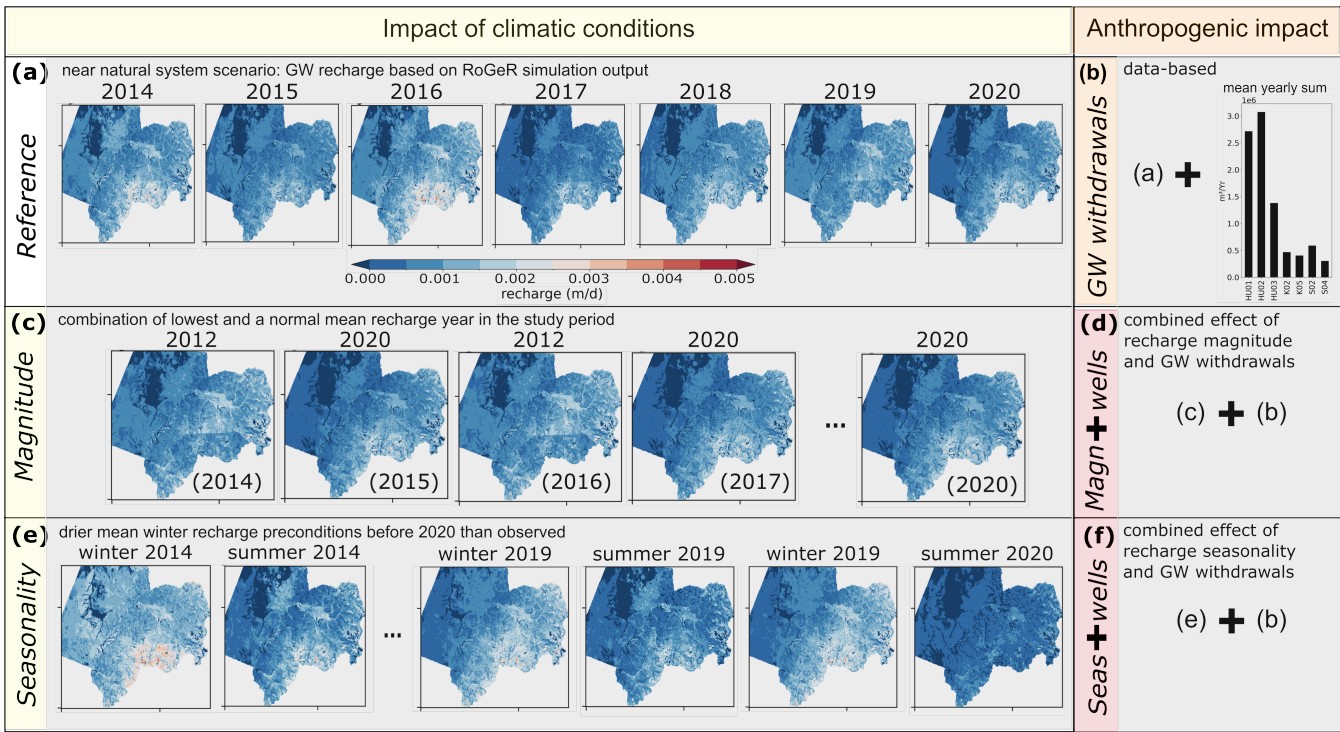

**Figure 2.** Concept showing all model experiments with a) the reference recharge input and c) and e) the modified input of the stress test scenarios for climatic (recharge) stress; and furthermore with b) the anthropogenic impact withdrawal scenario and d) and e) the combined effect. For climate impact, the annual mean recharge of the water year or the season are shown. For anthropogenic impact, the mean yearly sum of GW withdrawal rates are shown.





The water year with the lowest annual GW recharge is 2017, and the winter season with the lowest GW recharge is 2019 (Figures A6 and A7) (a water year starts in November with the summer season starting in May). We compare the stress test results (leakage) against a reference simulation, which is the near-natural system scenario (Figure 2 (a)). For the simulation of the near-natural system we neglect all water uses in the study area. Due to the long history of SW and GW water uses in the catchment (dams, weirs, small scale hydropower, straightening of water courses) and the lack of long-term data on the scale of our study area, we consider that an assumption of a natural system scenario is highly uncertain. Hence, we can only analyse potential connectivity changes due to specific stresses applied or removed from the system with respect to this near-natural system status.

## 3 Results

### 3.1 Validation of zero water levels

The ZWL occurences were measured at 20 locations in the study area (Herzog et al., 2022). A precise quantification of ZWL days at the scale of the stream reaches is not possible because GW levels are available for grid cells with $100\,\mathrm{m}$ resolution whereas streamreach units are smaller and may vary (reach width and length are declared in the parameterisation). A comparison of mean simulated water levels against observed water levels indicates, that the simulated water levels are underestimated despite the agreement of observed and simulated mean GW levels and relatively good agreement of observed and simulated streamflow (Figures A2, A1). In general, streamflow is slightly overestimated. This suggests, that the underestimation of water levels is likely due to the streambed parameterisation while the dynamics of the GW system and streamflow are correctly represented with the model. For convergence of measured and simulated ZWL days we performed a calibration of the threshold for the ZWL days based on the measured ZWL days from (Herzog et al., 2022). In general, the simulated percentage of ZWL days is overestimated (as expected due to the underestimation of the water levels) at all stations except for E2 and in particular for stations at very far distances from the main gauging station. The percentage of observed and simulated ZWL days is in good agreement for almost 50% of the stations (E4, W4, RO2A, E6, E8, E2), whereas differences in percentage are greater than 15 % for the other stations.





**Table 2.** Percentage of observed and calibrated simulated ZWL days for the study period ordered by distance to main gauging station (grey shaded rows with bold numbers depict a percentage difference < 15% between the observed and simulated ZWL days.)

| station | T | $T_{wells}$ | observed ZWL (%) | simulated ZWL (%) | simulated ZWL Wells (%) | distance PE (km) |
|---|---|---|---|---|---|---|
| **E8** | **0,9** | **0,0** | **74** | **89** | **81** | **0,5** |
| D3 | 5,7 | 5,3 | 15 | 43 | 38 | 1,5 |
| **E6** | **1,0** | **0,0** | **43** | **39** | **31** | **1,7** |
| RE1 | 0,9 | 0,9 | 3 | 34 | 28 | 1,9 |
| **E4** | **0,3** | **0,0** | **24** | **29** | **29** | **3,0** |
| D2 | 3,8 | 3,8 | 8 | 41 | 36 | 3,7 |
| **W4** | **5,3** | **5,3** | **31** | **41** | **36** | **5,3** |
| **E2** | **-0,1** | **-0,2** | **13** | **0** | **1** | **5,4** |
| **RO2A** | **0,7** | **0,7** | **55** | **60** | **57** | **6,2** |
| IB2 | 1,4 | 1,4 | 7 | 51 | 47 | 8,4 |
| W3 | 5,0 | 5,0 | 7 | 55 | 51 | 8,7 |
| W2 | 5,8 | 5,8 | 19 | 70 | 63 | 10,4 |
| IB1 | 0,2 | 0,2 | 13 | 90 | 58 | 12,1 |

## 3.2 Simulation of GW leakage

### 3.2.1 Spatial distribution of GW leakage in the study area during different GW conditions


The spatial distribution of leakage flow (L) differs for the driest GW conditions (days with highest mean depth to GW) and the wettest GW conditions (days with lowest, simulated mean depth to GW) in the simulation period (Figure 3). In general, depth to GW is lower in the north eastern part of the catchment. Strong topographic gradients towards the border of the catchment lead to strong hydraulic gradients which are problematic to represent as an average for one grid cell and therefore depth to GW

is less reliable in these areas. (Figure A3).

For dry GW conditions the leakage approaches zero especially in the northern part of the catchment (E tributary) (Figure 3 a) and b)). Including the wells in the simulation leads to stronger decrease in leakage in the downstream part of the E tributary (for all stress tests with wells). The recharge scenarios affect the timing of GW drought. The dry GW conditions occur earlier in summer and wet GW conditions earlier in january (at least for the magnitude recharge scenario) but the leakage pattern

itself for dry conditions and wet conditions does not differ significantly from reference/near-natural conditions (only if wells are included additionally). For wet GW conditions, the river system is entirely connected to the GW system in the study area as there is almost no place with zero leakage.

The difference in mean leakage ($\Delta$ L) of the reference conditions versus stress test conditions shows, that main differences in the downstream part of the catchment are caused by the implementation of wells in the simulation (Figure 3 c)). Concerning

the mangitude and seasonality recharge scenarios, the $\Delta$ L is slightly higher positive (leakage in natural system is greater) in





the upper part of the E tributary and in the downstream part of the eastern tributaries (towards W and R tributaries). As in Fig. 3 a), the effect of wells and modified recharge adds up when both are combined in one stress test. The standard deviation of leakage is highest along the E river, indicating that the variability of leakage is particularly high along this tributary (Figure 3 d)). Standard deviation is particularly low at the W and SW borders where slopes are increasing.

However, due to the abrupt change of simulated GW head at the catchment borders A3, the simulated leakage is highly uncertain for areas with increasing slopes. Nevertheless, the results give a general overview where in the study area leakage variability and dynamics are particularly strong and more specifically under which conditions they may change.

**Figure 3.** Spatial distribution of leakage during a) mean maximum depth to GW conditions, b) minimum depth to GW conditions, c) the relative difference between leakage of the stress tests and natural conditions d) the relative standard deviation of leakage for each of the scenarios.





### 3.2.2 Longitudinal variation of leakage flows

As leakage quantities cannot be validated but depend on simulated streamstage/streamflow, we analyse the longitudinal varia-

tion of GW leakage in detail for the different measurement locations in the following. We divided leakage flow by the simulated leakage flow area in $m^2$ to obtain specific leakage. For better visualisation and comparison, we normalized the leakage flow of 2020 (setting the minimum to zero. Minimum for the reference simulation and the well scenario are close). Quantities of leakage show strong variations in a longitudinal direction along the river bed (Figure 4) and a tendency for decreasing leakage variation with distance from the outlet (IB2, W3, W2, Za1). This pattern however has to be interpreted with caution, partic-

ularly due to the overestimation of ZWL days at the stations located far from the outlet. Characteristic for most locations in the main river D(D3,D2,D1) are loosing conditions (from a river perspective). Even though the main gauging station near the outlet (PE) is also located in the main river, it shows gaining conditions. Gaining conditions are otherwise primarily found

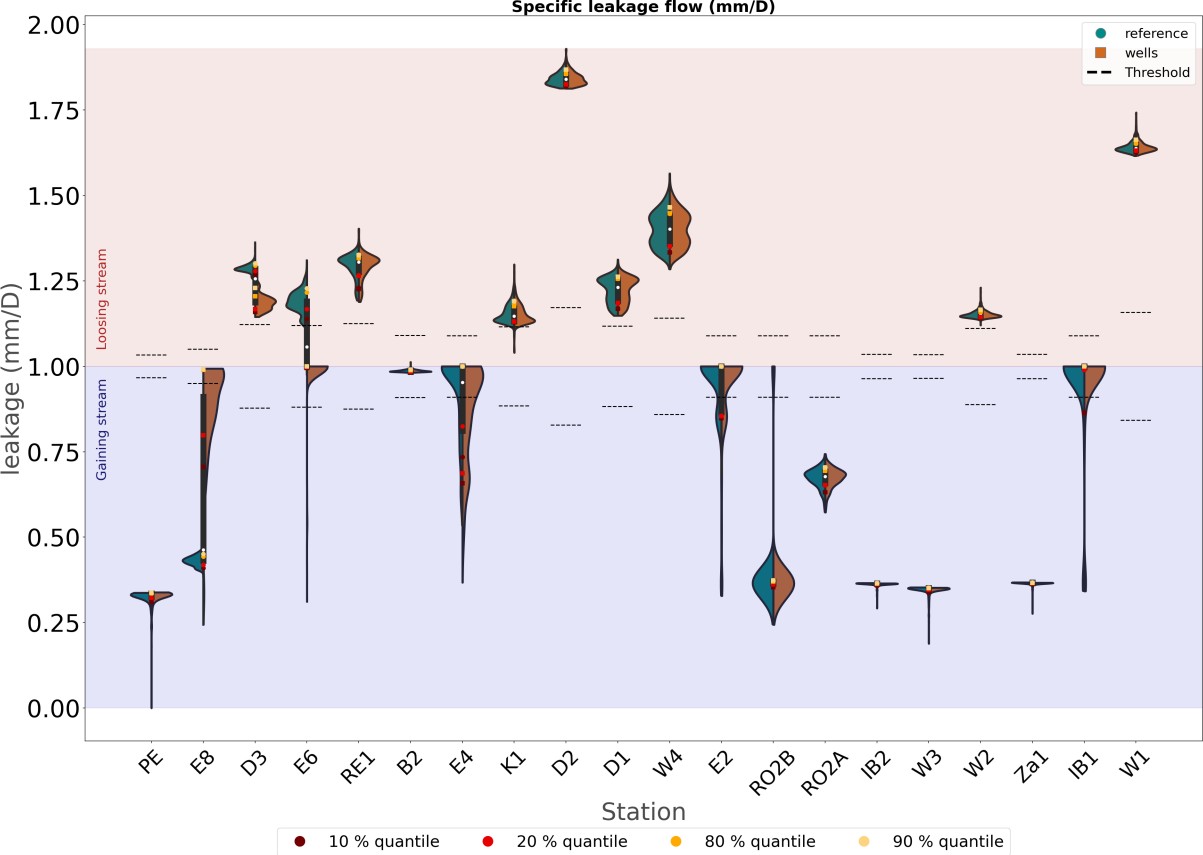

**Figure 4.** Normalized simulated flow between the aquifer and the riverbed for the reference scenario and the well scenario for the gauging station locations in 2020. From left to right the distance from the catchment outlet (station PE) is increasing. Values below 1 (above 1) indicate, that the water flow is directed towards the river (the aquifer).Note: Thresholds are location-specific as described in (see section 2.3).





along the E tributary (E8,E4,E2), except for E6 in the reference simulation. At most locations, there is no profound difference
between the reference and the well stress test. The most obvious exceptions are locations in the downstream E tributary (E6,
E8 and E4) as well as D3 in the D river.

### 3.2.3  Comparison of leakage direction changes

In order to understand which stream reaches experience changes from gaining conditions to no connection or loosing condi-
tions, we first evaluate the temporal evolution of direction changes at the specific locations for the different stress tests. Three
different GW leakage conditions may occur: zero leakage, meaning that there is no exchange between the river and the aquifer,
positive leakage (from the perspective of the GW body), meaning that the river experiences losing conditions and negative

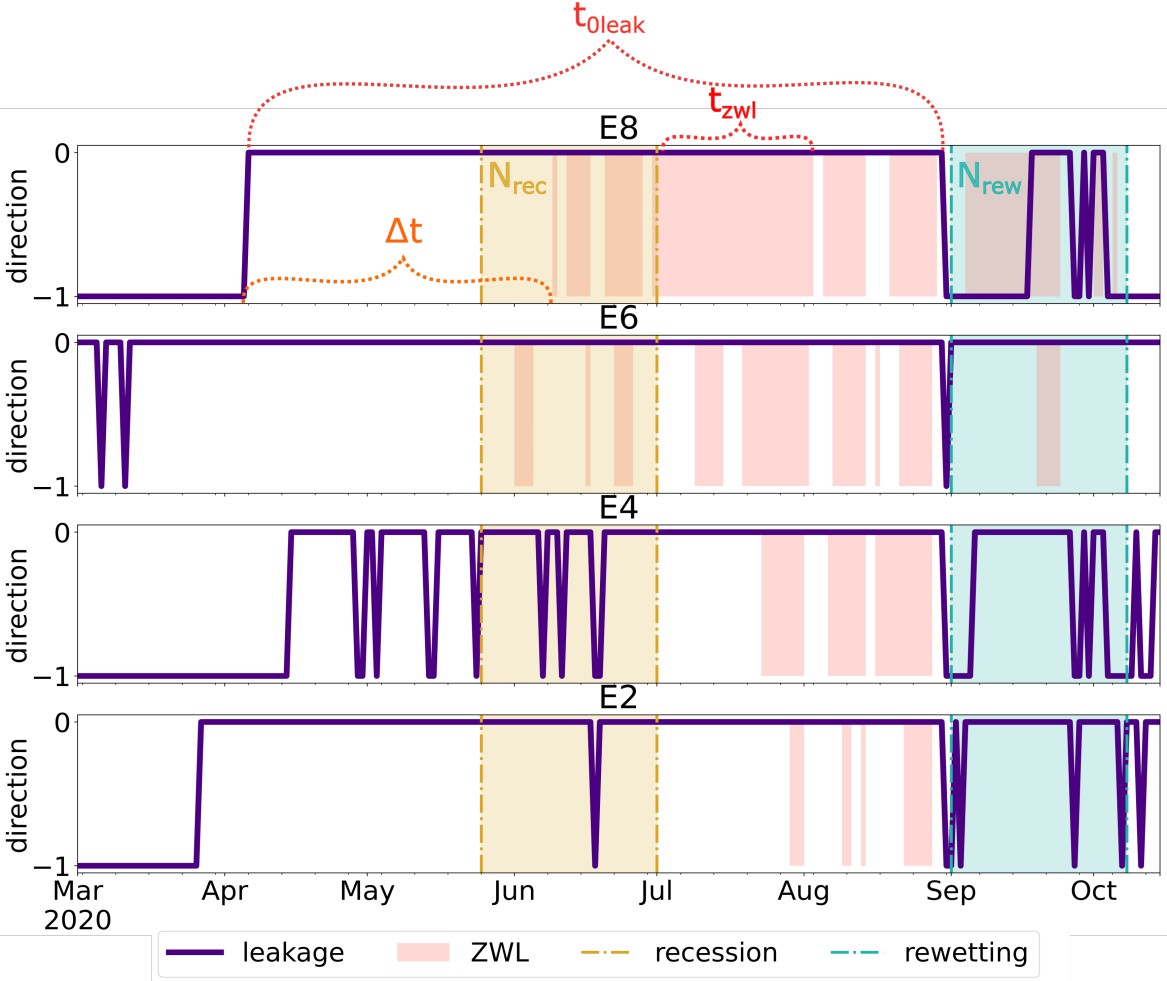

**Figure 5.** Leakage direction changes and ZWL days for the well scenario at the stations along the E tributary.





leakage, meaning that the river experiences gaining conditions. For normalized leakage (with the minimum equal to zero and zero equal to one), we define zero leakage within a range of $[1 - T_{leak}, 1 + T_{leak}]$, where $T_{leak}$ is the location specific threshold (as illustrated by the dashed lines in Fig. 4). For the following analysis we make a choice of locations based on the occurence of leakage direction changes firstly. Note however, that there are differences in how well ZWL days have been modeled at these

locations (Table 2). The characteristics of the temporal evolution of zero leakage and ZWL days form the metrics as described in section 2.3 (Figure 5). While E8, E4 and E2 experience only direction changes from gaining conditions to zero exchange, E6 experiences direction changes from losing conditions (for natural conditions) to mostly zero exchange (Figure A5). However, E6 stands out because it is the location closest to the wells with the highest abstraction rates and the flow direction changes differ significantly between stress tests with and without wells. In general, the upstream stations along the E tributary experience

less ZWL days than downstream locations.

### 3.3 The relationship of vertical and longitudinal connectivity

For duration $\delta t$, values above 0 (below 0) indicate, that the duration of the longest zero leakage event $t_{0leak}$ is greater (smaller) than the duration of the longest zero water level event $t_{zwl}$. For almost all stress tests, $t_{0leak}$ is greater than $t_{zwl}$, except for

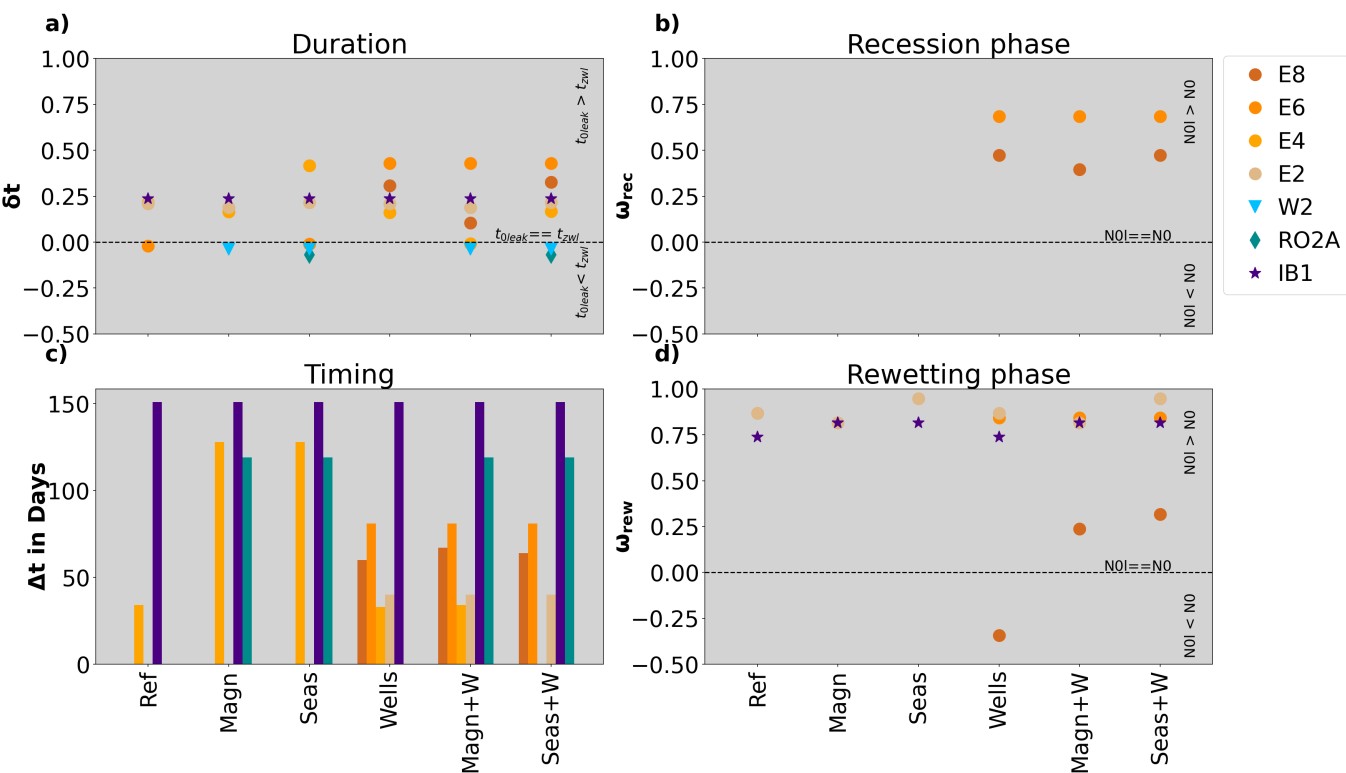

**Figure 6.** Results for duration and timing metrics for the different stress tests at all stations with zero leakage and ZWL days.





the regular dry-normal combination (sce1) (Figure 6 a)). For stress-tests without wells, the ratio is closer to 0 for E6 and W2.
For E8, no zero leakage was simulated in these scenarios. In all stress tests with wells $\delta t$ increases with zero leakage becoming
more important at E8 and E6. At E4, $\delta t$ is more heterogeneous, less influenced by well withdrawals but without a clear tendency
pointing towards a sensitivity to climatic conditions either. However, at E2, $\delta t$ does not vary significantly among scenarios. E2
is also the last station towards the upstream end of the tributary. At IB1, no difference is observed in comparison to the stress
tests without wells. $\delta t$ is more sensitive at W2 and RO2A to the recharge stress tests as zero leakage does not occur at these
locations in the reference simulation. Similarly at RO2A, where only dry winter preconditions make a difference.

Frequency $\omega$ differs only slightly for the recession and rewetting phase (Figure 6 b) and d)). For stress tests without wells,
almost no zero leakage occurs during the recession and rewetting phase. Hereby, it is problematic that observations don't show
ZWL days in the recession phase for some stations. However, for stations such as E8 and E6, where ZWL days occur in the
recession phase (Figure 5), zero leakage occurs far before it and thus, no $\omega$ can be defined. Nevertheless, some differences
can be analysed. At E6 $\omega$ has the same magnitude for the rewetting and recession phase. At E8, $\omega$ is significantly lower with
wells included for the rewetting phase in comparison to the recession phase pointing towards a stronger importance of ZWL
days in this phase (as leakage might switch back to gaining conditions earlier). However, the other stress tests with wells show
a higher $\omega$ during rewetting than during recession. The timing (i.e. time delay) between first zero leakage and first ZWL day
gives further information about whether ZWL occurs directly in response to zero leakage or not (Figure 6 c)). For scenarios
without wells, no zero leakage occured before ZWL in 2020 at all stations except E4 and IB1 in the natural system scenario
with $\Delta t$ being particular high at IB1. In sce1 and sce2, zero leakage occurs before ZWL at RO2A. However, in the scenarios
with wells $\Delta t$ increases for almost all stations (excluding W2). The presence of wells therefore seems to result in a time shift
of zero leakage appearing earlier than in the reference simulation and also earlier than ZWL. Interestingly, $\Delta t$ does not change
at E4 for the well stress tests (wells, s1w, s2w), but it increases for climatic stress tests. The latter points towards zero leakage
occuring earlier under dry recharge conditions. As for $\delta t$, $\Delta t$ at RO2A is only changed by recharge stress tests but the magnitude
of $\Delta t$ does not change among them. At W2 no result is found for $\Delta t$ as there is rarely any zero leakage found (Figure A5).

## 4  Discussion

### 4.1  Simulation of drying dynamics

The first research question addressed to what degree the model setup would be able to simulate the dynamics or patterns
of connectivity and consequentially the river bed drying in the Dreisam and its tributaries. Overall, the model did simulate
ZWL days well for locations in the valley bottom. An analysis of patterns of longitudinal and vertical connectivity based on the
model results allowed to distinguish between predominantly gaining and losing stream reaches in general. The simulated spatial
patterns of connectivity depicted where reversals of connectivity are likely to occur for different GW conditions. Nevertheless,
differences between observed and simulated ZWL days are substantial at some locations. Possible reasons of these differences
likely relate to several simplifications in the models, both the modelled recharge input as well as the groundwater model
and how ZWL are derived from it. Most important to name are the conversion of GW heads from the scale of a grid cell





to a river reach unit and the uncertainties in riverbed parameterisation. Other studies have found such difficulties before. (Brunner et al., 2010) showed, that coarse resolutions of the Modflow GW model can lead to underestimation of infiltration in losing streams, which results in lower GW head and thus also influences streamflow results. Increasing model resolution

may reduce this problem (Mehl and Hill, 2010). Strong topographic gradients and coarse resolution do not only impact GW heads but also result in distortion of hydrogeologic properties (Foster et al., 2020; Fleckenstein et al., 2006). This could also affect simulated hillslope contributions, which this study did not focus on with high detail. If hillslope contributions were not correctly represented, this could have also affected modeled ZWL at some upstream locations (for example IB1). Additional observations on hillslope connectivity would be necessary to identify where the model misses such inflows into the main

tributaries from the headwaters or the hillslopes. In general, the influence of spatial resolution on modeled water levels and streamflow (e.g. by decreasing the GW grid cell size) should be further assessed in future studies in order to use this model for a quantification of surface water availability.

The riverbed parameterisation depends on available raster data and human structures such as weirs and bridges have not been considered in the parameterisation. Furthermore, the spatial discretisation of stream reaches likely does not represent the small-

scale heterogeneity of streambed parameters observed in reality. Hence, the heterogeneity of the dynamics at specific stations may not be captured due to this reach-scale homogenisation of streambed parameters in the model. In order to estimate which streambed parameter has the largest influence on the model results, a sensitivity analysis would be meaningful to understand which streambed parameters have the largest impact on the model results. Another option to reduce parameterisation errors is to calibrate hydrodynamic parameters based on hydrodynamic models (Quan et al., 2020) or conceptual models (Meert et al.,

2018; Vermuyten et al., 2018) instead of relying on available gridded data.

The objective of this study was to obtain a more general overview on the sensitivity of the spatiotemporal dynamics of GW-SW-interaction towards different environmental stresses, which did not require the best absolute simulation result with respect to groundwater levels for example, but rather a representation of the correct responses to changed input or conditions in relative terms. The aim was therefore to reduce calculation times, which is the main advantage of offline-coupled models (Condon et al.,

2021). A prerequisite for future approaches to quantifiy leakage flows in the study area is the availability of validation data and further development of the model presented in this study. With respect to this, a comparison of the presented model approach against other model structures (especially considering fully coupled models) for the simulation of GW-SW-interaction could also be envisioned.

## 4.2 Appraisal of connectivity metrics

The second research question addressed an important gap of metrics that may help compare modelled and measured vertical and longitudinal connectivities. The metrics as described in Table 1 allow a comparison of the temporal dynamics of vertical and longitudinal connectivty at different locations in the catchment. The joint analysis of longitudinal and vertical connectivity confirm the impact of water withdrawals in the north-eastern part of the catchment as in Fig. 3 but they also allow to demarcate locations that are likely more sensitive to climatic impacts. The investigation of $\delta$ t shows, that zero leakage persists for longer

durations than ZWL for almost all stations (except the ones experiencing very few zero leakage), supporting that ZWL days do





not occur independently of zero leakage. $\delta$ t, $\Delta$ t and $\omega$ along the downstream E tributary stations are influenced by the presence of wells. RO2A and IB1 are most influenced by climatic preconditions, which was shown to have an impact on $\delta$ t (indicating that zero leakage appears) and $\Delta$ t (only for RO2A) but not on $\omega$ on the other hand. This different behavior underlines that the relationship of vertical and longitudinal connectivity is very site specific. W2 and IB1 are both relatively far from the outlet but

IB2 is experiencing a lot and W2 very few zero leakage. These differences can only be expressed looking at multiple temporal characteristics, which highlights the value of the use of such metrics. In this study, zero leakage was mostly found at stations characterised by predominantly gaining conditions (see also Figure A5). The findings of this study are therefore not necessarily valid for the spatio-temporal relationship between vertical and longitudinal connectivity in losing stream sections. Differences among losing and gaining streams in their connectivity relationship therefore requires more research in different catchment

context and with larger samples.

Additionally, one also has to be aware of the limits of the metrics due to their definition. First of all, the metrics can only be used for streams experiencing dry spells and zero leakage. The metric for timing could not be calculated if zero leakage does not occur in the same year (stations without bars in Fig. 6 c)). Considering longer time spans, infinite or very large values would appear for location where the time delay between changes in vertical connectivity and longitudinal connectivity is large. But

such large values are difficult to display and imply that there is no link between vertical and longitudinal connectivity changes anyway. Furthermore, the frequency metric is difficult to assess when looking at one year only. For a long-term analysis of the evolution of connectivity changes, the frequency metric would serve to compare the recession and rewetting phase in different years. However, this is problematic for analysing intraseasonal differences. The definition of recession and rewetting phase depends on available data (May-November 2020) and same number of days in each phase but could also be defined differently

to actually take into account, that each station shows a different drying and rewetting pattern, which is not always happening at the same time during the year. A possibility to adjust the frequency metric to intraseasonal timescales would be to extract the actual recession period at each station based on the hydrograph.

In general, the focus of this study was on understanding the spatio-temporal relationship of leakage flow and ZWL. We neglect the metrics describing seasonality due to the focus on the summer season. We also neglect metrics based on leakage quantities

(magnitude, rate of change) due to the difficulty to combine counts (zero spells) with a magnitude (amount of leakage) and because we are not able to validate leakage quantities without leakage measurements. In the following, metrics focusing on the relationship of quantities of leakage flow and ZWL at different locations should also be developed. An analysis of leakage flow quantities however requires more work to obtain the best simulation result and validation data for leakage. Only zero leakage and ZWL were in focus. Additionally, different leakage directions could be related to ZWL or also to zero flow. This would

help to describe the leakage conditions at each station in general (whether gaining or losing conditions are dominant).

### 4.3 Sensitivity of leakage flow to stress tests

The third research question addressed the sensitivities and changes of modelled connectivities in response to the applied stress test scenarios. Following the findings of (Hellwig et al., 2021), who found that baseflow reacts on shorter timescales to intensified drought events (especially in fast reacting systems), one would have expected drier recharge preconditions to





modify leakage flow. Surprisingly, leakage of the climatic stress tests did neither vary significantly for dry nor for wet GW conditions compared to the reference simulation (Figure 3). Leakage variation was highest for dry GW conditions for stress tests with water uses although the largest variability of leakage was focused on a specific part of the catchment. This suggests, that anthropogenic activities such as water uses induce fluctuations of GW heads at short timescales and may therefore locally have a larger influence on GW-SW-interaction on such short time scales than climatic conditions. Particularly, this concerns the

occurrence of zero leakage in specific parts of the stream network. Hereby, the critical distance to the wells should be further investigated, for example by means of using stream proximity criteria (Zipper et al., 2019; Li et al., 2022). In our simulations, the well locations and depths were based on observations. However, synthetic stress tests with equally distributed wells and different withdrawal rates could also be envisioned to obtain a general understanding of the impact of water withdrawals. This could help to answer practical questions, such as where GW contributions may buffer the effects of well withdrawals or

for the definition of critical withdrawal rates. Critical withdrawal rates could also be investigated, taking seasonality of water withdrawals into account, for example to compare stress tests with more and no water withdrawals in summer.

The stronger impact of well withdrawals on vertical connectivity in comparison to recharge stress may not be valid for larger timescales as particularly GW is a slow reacting system (Cuthbert et al., 2019). One has to note also, that we designed the stress tests to account for current climatic trends of dry recharge years and dry recharge winters to occur more often and not

for climatic extremes. We also did not evaluate the effect of longer durations (more than two years) of dry recharge conditions, which have been shown to increase duration of streamflow droughts in porous aquifers (Stoelzle et al., 2014). In general, the interpretation of changes induced by the magnitude of the climatic stress is more complex in comparison to the seasonality stress test because the whole recharge sequence was modified.

By evaluating leakage only for the driest and the wettest GW conditions in the study period, we obtain an overview on how

leakage differs in the extreme (GW) situation. While this may be particularly relevant for water management, the understanding of how vertical connectivity evolves in response to GW could be enhanced by an additional event based analysis (e.g. looking at whole periods of relatively deep (or low) GW head) or an analysis of the temporal evolution of leakage during changing GW conditions. These synergies between connectivity changes and drought imply furthermore, that research on IRES and on drought should not be independent, as elaborated e.g. by (Yildirim and Aksoy, 2022).

While the influence of climatic stress tests on leakage was low, they obviously had an impact on the timing of the occurrence of dry GW conditions. With dry winter preconditions in 2020, the driest GW conditions occurred in late august 2020 and with every second year being the driest modeled recharge year between 2009 and 2020, driest GW conditions occurred in late august 2017, as compared to late august 2018 in the reference simulation. On an interannual basis, GW heads were generally lowest in the end of summer, which is in agreement with trends found in other studies based on climate scenarios (Dams et al.,

2012). This shift of dry GW conditions due to climatic stress could also lead to a shift in timing of zero leakage in general. However, the timing of dry GW conditions could also be caused by the interplay of different hydrologic variables in specific years, leading to more or less resilience of the GW system to climatic influences. To assess this further was not the central interest in this study due to the focus on the interannual changes in the year of 2020. But a broad understanding on vertical



connectivity changes due to climatic stress can likely only be achieved through a multi annual analysis of the effects of climatic
stress tests.

## 5  Conclusions

This study represents a very general, first attempt to model GW-SW interaction in the Dreisam valley, which helped to gain
insight into factors to consider in future modeling studies. The evaluation of leakage flow for dry and wet GW conditions
identifies parts of the catchment, with particularly strong variability of GW-SW-interaction and where GW-SW interaction
ceases during dry conditions. This was found for natural conditions and all the stress tests. However it has to be kept in mind
that 'natural conditions' with no abstractions and an untrained river system have not existed in this catchment for more than
50 years. Hence, the reference cannot be validated. Furthermore, uncertainties in modeling leakage flows are still high. Firstly,
potential inaccuracies in the streambed parameterisation are passed to leakage flows due to the dependency of simulated
leakage on simulated water levels; Secondly, quantities of simulated leakage flow could not be evaluated due to the lack
of validation data on GW-SW interaction. Due to the uncertainty of the model, we were not able to investigate changes in
magnitude of connectivity. Additional simulations (with particular focus on parametrisation and resolution) and validation data
for leakage flows and hillslope contributions are required to actually quantify the impact of different stresses on connectivity.
However, an analysis of metrics describing the relationship of zero leakage and ZWL (duration, timing and frequency) helped
to disentangle the spatio-temporal relationship of zero leakage and ZWL at specific locations. The stress tests showed, that
well withdrawals might significantly affect leakage in the same tributary. We showed that well withdrawals influence the
intraseasonal relationship of longitudinal and vertical connectivity (duration, timing and frequency), in a way that zero leakage
becomes more dominant and thus, vertical connectivity decreases. The magnitude of the decrease in vertical connectivity is
possibly influenced by the distance to the wells but in general, our analysis showed that connectivity fluctuations increase and
are exacerbated during the recession and rewetting phase in a specific part of the stream network. However, the combined
analysis of vertical and longitudinal metrics reveals, that the distance from the wells is not the only factor leading to zero
leakage. For some locations (particularly upstream ones), climatic preconditions were the primary influencing factor. This
indicates furthermore, that changes of connectivity patterns in response to different types of stresses might differ depending
on the location (upstream or downstream). This could be a starting point for future analysis of such connectivity differences
between upstream and downstream locations. Future analysis should also focus on zero flow in addition to ZWL and not only
on zero leakage but on all types of leakage directions and on multi-annual analysis.

*Code and data availability.* The Zero water level dataset is publically available (Herzog et al., 2021b). The streamflow dataset will be
published.



*Author contributions.* Amelie Herzog was responsible for running the simulations, data treatment of the model outputs, the design and analysis of the study as well as all figures and writing of the manuscript. She also provided data for validation of the model (zero water levels and streamflow). Jost Hellwig gathered the input data for the modflow model and performed initial runs with a prior Modflow5 version of the final Modflow6 model used in this study. Jost Hellwig and Kerstin Stahl both helped to finalise the manuscript.

*Competing interests.* The authors declare no conflict of interest.

*Financial support.* This work was supported by the Badenova Fund for Innovation

*Acknowledgements.* We acknowledge the work of Max Schmitt (and Hannes Leistert) for performing the RoGeR model runs used as input data for the Modflow6 model.





## Appendix A

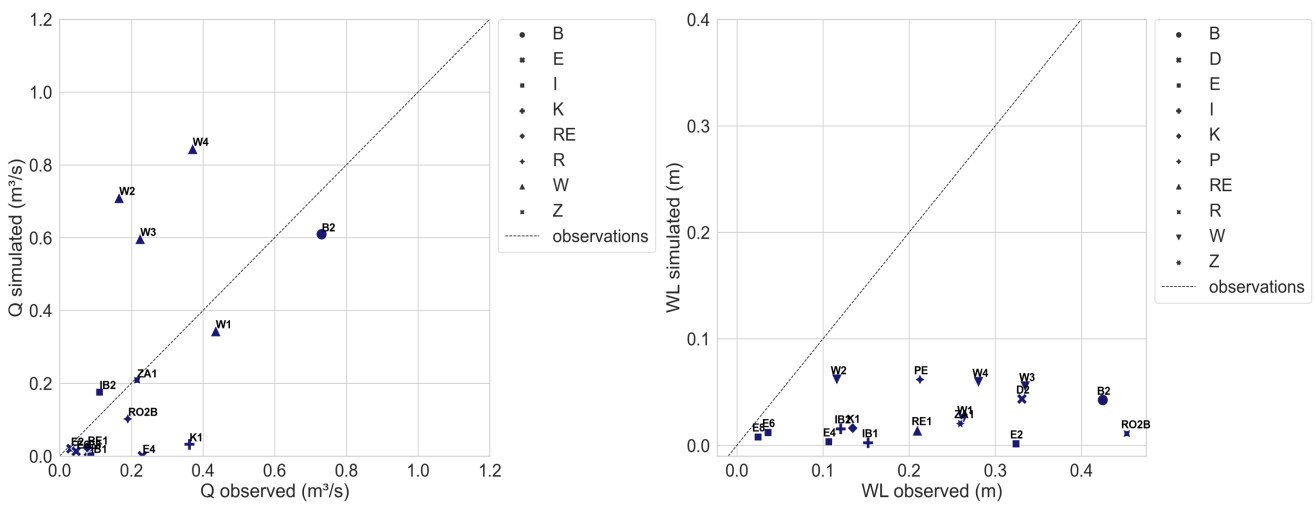

**Figure A1.** The simulated and observed streamflow and water levels in summer 2020

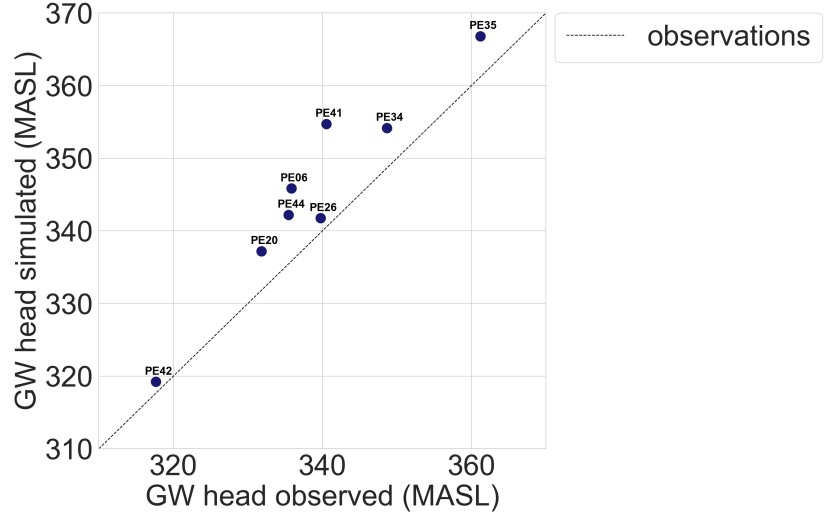

**Figure A2.** The simulated and observed GW head in summer 2020





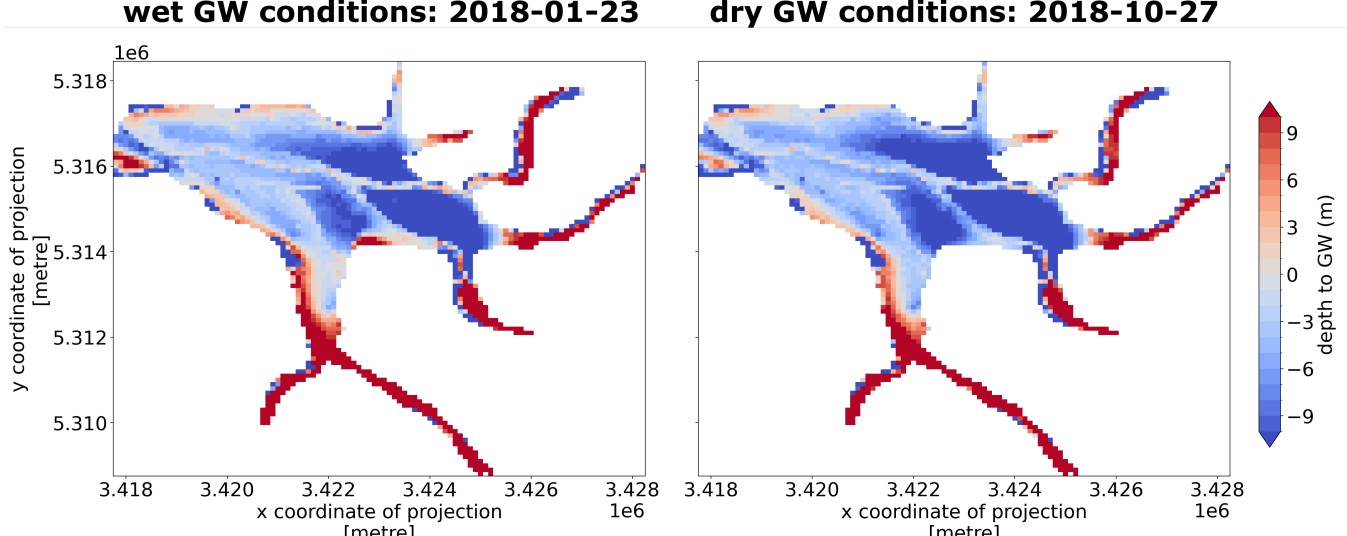

**Figure A3.** The simulated depth to GW for wettest GW conditions and driest GW conditions in the study period

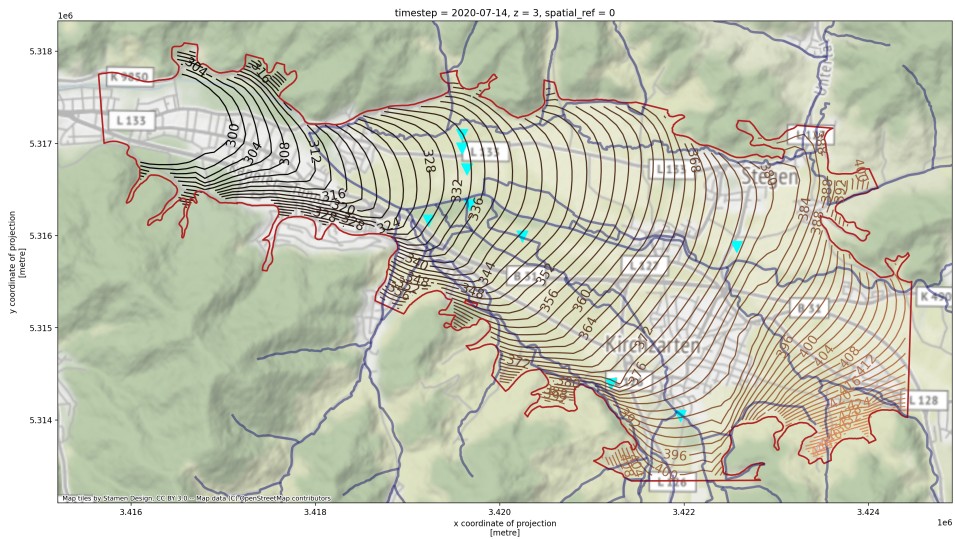

**Figure A4.** Simulated GW contour lines for the reference simulation for 14th of july 2020.





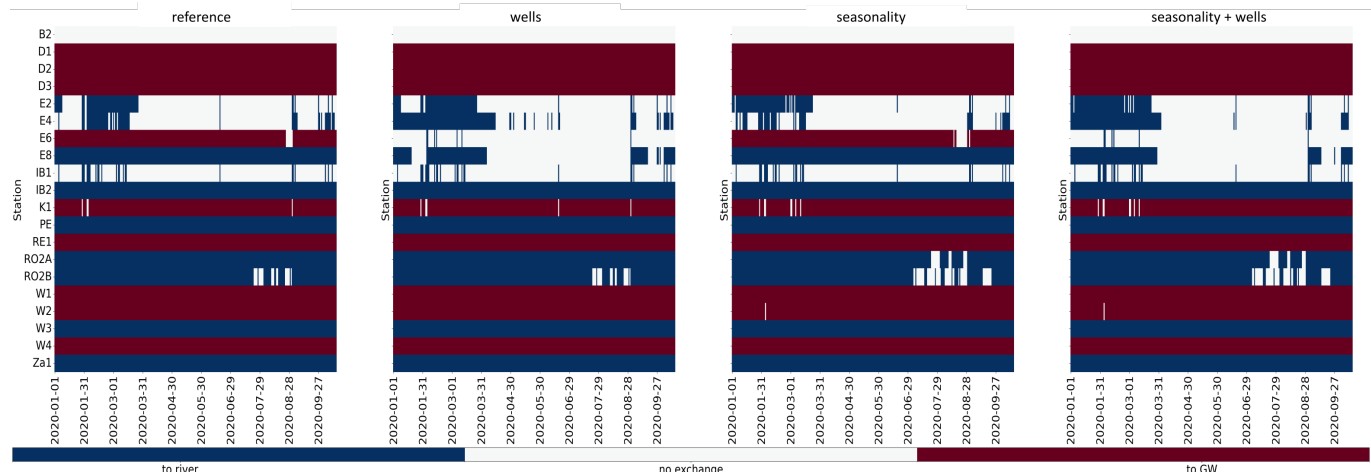

**Figure A5.** The extracted, normalized direction changes in 2020.

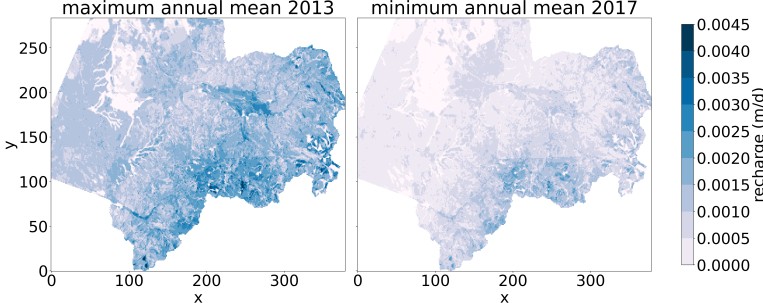

**Figure A6.** Mean annual minimum and maximum recharge water years for the period between 2009 and 2020 (RoGeR simulations)





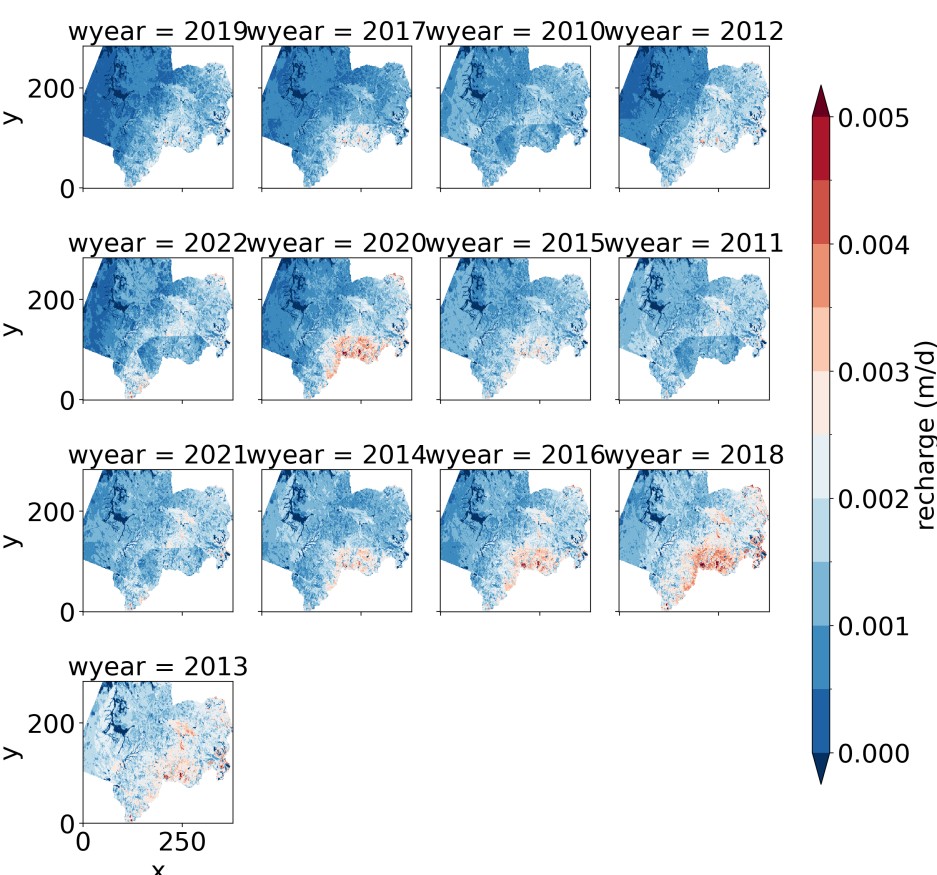

**Figure A7.** Mean annual recharge for the winter season between 2010 and 2020 (RoGeR simulations)





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
