# Peer review of "An investigation of anthropogenic influences on hydrologic connectivity using stress tests"

_Hydrology and Earth System Sciences, 2023_

## Author Comment (AC1)

**Reply to Referee 1**

Answers to the comments in blue

Herzog et al. Study the longitudinal, vertical, and lateral connectivity of GW and SW in the Dreisam valley. The study is nicely done, generally well described and scientifically interesting. Mainly, I suggest adding some more explanation and discussion to better guide the reader and thus better help her to understand the storyline presented. I generally like the study and have no major comments beyond the constructive suggestions made below.

We thank referee 1 for the positive feedback and the constructive comments below.

[1] There are a few editorial changes needed regarding the English. An example from the abstract: "This raises the question on" should be "question of". Or "By reason of the physically-based" would better read as "Due to the phy…". Please have another read through the document for these instances.

The manuscript will be revised with respect to the English language

[2] In the abstract, "model reality" should probably be "model realism" to be more in line with the language used in other papers (e.g. Gharari et al. 2014 HESS; Hrachowitz et al. 2014 WRR; Wagener 2003 HP).

Will be revised accordingly.

[3] I like the abstract, but could you quantify terms a bit more. E.g. what is a short time scale in the context of this study? How many measurement locations did you consider? Etc. This is all discussed in the manuscript but might make the contributions of the study clearer right away if mentioned in the abstract.

The term "short timescales" mainly relates to the model simulation time period (2014-2022). Therefore, it is not possible to assess long-term changes in GW dynamics due to GW withdrawals, which already took place before 2014. We will replace the term "short timescales" by the term "short time periods".

Regarding the measurement locations, the model is evaluated based on zero water level measurements at 20 locations in the Dreisam valley. However, for the evaluation of the relationship of longitudinal and vertical connectivity, a prerequisite is that locations experience both, ZWL days and zero leakage (7 locations). This is explained in the text (l. 273-275). These details will be revised in the abstract.

[4] It might be interesting to connect the metrics discussed, developed, and estimated in this study to the metrics (often called signatures) used in other studies. E.g. the recent study on reservoir impact in the UK by Salwey et al. (2023, WRR).

On the one hand, the approach to develop signatures and describe changes in these signatures due to water uses, (for example for water reservoirs in Salwey et al.) is somewhat similar. However, as in most other existing studies, these signatures/metrics are defined based on streamflow data only, which is similar to other classical approaches, such as the Hydrological Alterations approach. Such approaches help to identify changes in the streamflow regime and

dynamics but these approaches only consider longitudinal connectivity. In distinction, the objective in this study was to propose metrics, which include information on vertical connectivity as well to better understand the drying dynamics. The metrics consider dry phases (ZWL days) and connectivity changes (zero leakage, i.e. vertical connectivity). Thus, the signatures in Salwey et al. and the metrics used in this study are not comparable. We suggest that we will incorporate a more detailed discussion on the differences of our approach and existing signatures in the discussion section.

[5] The authors state in lines 69ff: "While such parameter uncertainties are relevant when it comes to obtaining the best model results, they are less relevant if the focus is on process understanding." I do not agree with this statement. Understanding which parameter dominate system responses, and what preferred values they take when they do so, has long been part of assessing models regarding their physical realism (e.g. Reusser et al., 2009, HESS). So dismissing parameter uncertainty as a simple problem of model performance is really understating the problem. I therefore would expect a discussion of the potential influence of parameter uncertainty on the study outcomes in the conclusions or discussions sections. Even if a more detailed analysis is not feasible in this study.

We agree that the sentence is too simplified and will expand our discussion on parameter uncertainty.

Late in the paper, the authors stated "The third research question addressed the sensitivities and changes of modelled connectivities in response to the applied stress test scenarios." Is this question really completely unrelated to parameter uncertainty? I can accept if the authors cannot add this element to this study, but a basic discussion of the potential influence would be good.

We agree that this research question is not completely unrelated to parameter uncertainty because the parameter uncertainty affects modelled ZWL days and connectivities. However, in this analysis we focus on the potential of model stress-tests and did not perform a sensitivity analysis in the way this term is often used in modelling terminology, i.e. sensitivity to varying parameter values. Responding to the comment, we suggest to weigh the potentials and limits of the approach in more detail and discuss how parameter uncertainty might affect stress-test findings.

[6] I am afraid that I am a bit lost when looking at Figure 5. The super short caption is hardly helping me to understand what I am looking at here. The text discusses gaining and loosing conditions. Maybe making those explicit in the figure would be a start? More info please.

The caption text will be adapted to be more explicit to express, that we are looking at examples of measurement locations which experience a direction change from gaining conditions (-1) to zero leakage (0).

[7] A general comment after looking at the next figure. Can you please make the captions more extensive. It is a bit annoying to have to go through much of the paper to look for abbreviations, variable names, location details etc. to understand figures. Please make the captions much more detailed so that the reader does not have to go through the text to understand the figure content. Or at least tell the reader exactly where to find the info needed to interpret the figure.

Captions (or where possible preferably the legends) will be expanded and improved.

[8] Regarding the conclusions. I understand that the authors discuss what they specifically learn about their study region. However, it might be nice to add a short paragraph on what innovations, understanding or questions might be transferrable to other studies. What outcomes are general?

Thanks for pointing this out. In the revised version we will work out transferrable knowledge gain such as

- the introduction and usability of connectivity metrics in different hydrogeological context and for different seasonality of the drying
- the potential to distinguish between climatic and human impacts on streambed drying using model stress test approaches
- the limitations of model approaches to simulate specific aspects of groundwater-surface water interaction

[9] As future work, the authors might want to consider a broader sensitivity analysis which could include both the stress test to the system as well as uncertainty in parameters or other model inputs. That would create a generic framework for analyses of the type presented here.

Thanks for this comment. Indeed, this would be the ideal follow up and we can add this to the outlook on future work.

---

## Author Comment (AC2)

**Reply to Referee 2**

Answers to the comments in blue.

I have read this discussion paper with great interest. I do not doubt the science that went into this study. However, the way it is presented is rather unsatisfying. This feeling covers all aspects of the paper, specifically how the methodology and results are presented as well as the interpretation of results and conclusion.

We thank referee 2 for the feedback on our study. We acknowledge the need for improvement of the presentation of the methods, in particular a more detailed description of the model and model setup, and of the results. We suggest details below (and also in response to R1). .

Overall I am a bit puzzled about how I should interpret this study. It starts off with a larger context. But, much more than a 'very general, first attempt to model SW-GW interactions in the Dreisam valley (L422-424)' it is not. I am questioning how much we learned 'to help gain insights into factors to consider in future modeling studies' (L424). What are these insights specifically and to what extent are the findings from this study transferable to other studies? (and is that not already done, for example, in the larger scale groundwater-surface water models available?).

We see that the phrasing of aims and insights and the discussion of transferability of the results are not clearly aligned. In the revised version we will clarify the local and general addressed knowledge gaps more clearly and better work out transferrable knowledge gain such as

- the introduction and usability of connectivity metrics in different hydrogeological contexts and for different seasonalities of the drying
- the potential to distinguish between climatic and human impacts on streambed drying using model stress test approaches
- the limitations of model approaches to simulate specific aspects of groundwater-surface water interaction

I regret there is no proper sensitivity analysis done of at least the most important parameter settings that impact the groundwater-surface water interactions. However, I do understand that this is probably not something the authors want to add to their analysis at this stage. I found all conclusions on uncertainties of the simulated leakage, because of the very limited model evaluation, somewhat hard to interpret. I have several points of concern:

Thank you for this comment. As already noted in the response to R1, we acknowledge that we used the term sensitivity not in the usual hydrological modelling terminology of a formalized (parameter) sensitivity analysis. We suggest we can change and clarify this in a revised version and respond to each of the specific points separately.

[1] **Model concept**. There are several aspects that I cannot fully understand. (L118) 'We used a combination of..'; I am not sure what this means. Are the models coupled in a way, or does one model use the outputs of the other, or did you use a combination of model results of the analysis? Not clear.

The Modflow model uses recharge and runoff data simulated with the RoGeR model as an input (see L129-131). Thus, we used an offline coupled model approach here. For clarification, we will specifically mention this in section 2.2.

(L123-124) 'Surface water fluxes, well extractions, and recharge are added in the form of boundary conditions'. How does this work in Modflow6? As far as I know, you have to translate surface water fluxes to heads and use it as a general head boundary in Modflow. To be able to calculate heads from fluxes there needs to be some calculations using assumptions on river depth and width. What was used here? For wells in Modflow, you can define this as a specific pumping rate or specific head. Recharge is probably used as an upper boundary and as an input flux. Is capillary rise considered as well? Not enough information is provided to understand how the Modflow model is built and how boundary conditions are implemented.

In Modflow6, the surface water fluxes, well extractions and recharge are added by means of the SFR-package, the well package and the RCH packages. Mathematically, the so called stress packages are boundary conditions (see Modflow6 documentation Langevin et al., 2017 chapter 6). In order to be more precise on this, we will specifically mention the packages used and refer to the Modflow documentation.

Capillary rise is not considered in Modflow but taken into account during the recharge calculation with RoGeR. In case of capillary rise recharge as the input flux for Modflow can also become negative

River depth and width are given in the streambed parameterisation. The data availability of these parameters is mentioned in L139/140.

The well package used in this study uses a flow rate/pumping rate for each well (see Langevin et al., 2017 chapter 6). Data for each well was available from the regional water supplier (see L110/111). We will add all this additional model setup information to section 2.2.

(L129-136): 'the model' is this referring to the smaller Dreisam model or the bigger model? 'requires spatially distributed parameters determining' what kind of parameters? Determining surface and subsurface flows so I am assuming soil parameterization, elevation etc but this is not mentioned anywhere. (L134): as the aquifers are unconfined (at least I did not read otherwise) does it make sense to use specific storage (neglectable of unconfined conditions) compared to specific yields? (L156-157) 'based on Manning': I think this needs a bit more explanation (or reference to the original code and/or coupling between surface and groundwater).

As the extension of the model is bigger than the Dreisam catchment (see previous L127-129), model input is required for the entire model domain. L129/130 should actually read: "The model requires spatially distributed parameters determining surface and subsurface flows as well as timeseries of runoff and recharge as an upper boundary condition" We regret this mistake and we will change the phrasing. The following lines (L130-135) explain the input parameters in detail. We will add information on topography, which is available from a DEM of approximately 30 m resolution.

We mentioned unconfined conditions in L126 but we will change the phrasing to make clear that we assume unconfined conditions in the model. For the simulation of GW flows, Modflow requires information on specific storage (L134-135) and specific yield. The

streamflow routing uses a continuity approach (water budget) and thus, Manning's equation is only used for conversion of streamflow to stream depths (L151-158).

(L142-158) A general remark for this whole section is that there is not a clear structure, especially not in this last part. It might help to show a small model conceptualization or flow chart: what goes in (parameterization, forcing) and what comes out (discharge, which is then used to calculate river heads), etc. I truly recommend going through this section again and making sure everything is very clear: which input data and where did you get it from, what is calculated in which model component, and how does it feed into the next model component? A proper understanding of the model setup is crucial to understanding and interpreting results as a reader (also, specifically as no sensitivity analysis is performed on any parameter settings. Then at least be clear and open on how you construct and parameterize your models).

We agree and think we can improve the clarity of model setup and workflow as part of a revision, possibly considering an additional schematic to the existing figure or it's improvement. Here we can point to the building blocks that are already in the text and that we can easily expand on in a revised version: The input and output data are mentioned in section 2.1. A conceptual figure of the model input and output is given in Figure 1. For the information about how river heads are computed from discharge we refer to L151-158 and the Modflow6 documentation (Langevin et al., 2017).

Some additional questions: the RoGeR model was at daily resolution, how about the groundwater model and the river routing? Same for spatial resolution? And vertical discretization of both the hydrological and groundwater models is also not that clear to me. Human interactions are implemented by groundwater extraction only. And is there a return flow to surface water and/or groundwater I cannot follow that either.

The spatial resolution of the GW model is 100x100m (see L129). RoGeR's spatial resolution is much higher as it runs separately for homogeneous polygons regarding elevation and soil parameters. Recharge and runoff for all intersecting polygons are summed for each grid cell to generate Modflow input data. We will add the missing information to the revised manuscript.

Modflow's vertical discretization comprises four layers. In L134, we describe the subsurface parameterisation data and mention the data source. Figure 1c) contains a cross-section of hydraulic conductivities and the vertical discretization along this cross-section.

In this study, GW extractions are the only human influences considered. We agree that there are other human influences, such as urbanisation, soil sealing, land use changes, or water withdrawals due to irrigation, which can also influence recharge, groundwater heads and interaction of groundwater and surface water. However, in our study catchment of river Dreisam groundwater extractions for drinking water supply is by far the most important human influence. To assess other factors, model stress test approaches, such as presented in this study, can be adapted accordingly in the future. For example, the Modflow6 drain package can also be used to implement agricultural drains and other stresses, which potentially modify the GW head. We will add discussion on this as an outlook in the discussion section.

Groundwater extractions are used for drinking water supply of the city of Freiburg, which is located downstream of the study area. Sewage plants for Freiburg are located downstream of our study area as well. Therefore, we can assume that there is no relevant proportion of return flow within our study area.

(L142-150) More generally, I have some trouble understanding the term 'leakage' in this study. As I understand it now, it is used here to describe groundwater drainage and river infiltration. For me, leakage would mean more the unintended movement or loss of groundwater from its natural subsurface flow and not the dynamic interaction between groundwater and surface water that changes due to groundwater pumping. Maybe the terminology is something to verify with a groundwater expert as well.

Thank you for this comment. The objective of this study was to simulate GW-SW interaction. In this study, leakage therefore refers to the exchange flow between GW and SW in this study. We agree that the use of the term "leakage" is probably not ideal. However, we used the term here for consistency with the Modflow documentation. If we keep the term, we will clarify this better.

[2] **Evaluation of model results**. Similar to the previous I have many questions. More in general I find it hard to understand what ZWL represents. A conceptual figure just showing a connected groundwater-surface water system and a disconnected system might already help.

We regret that the definition of ZWL is not clear. ZWL means, that the streambed is dry. This means, that an interruption of longitudinal connectivity takes place once ZWL occurs in a specific stream section. We will add this information in the beginning of the manuscript.

The direction of GW-SW-interaction depends on the relationship of vertical (leakage direction) and longitudinal connectivity (ZWL). We did not provide a conceptual figure here as we explained this in L182-186. The relationship of leakage direction and ZWL is additionally displayed in Figure 5. We will revise the description and legend of this figure to make this more clear to the reader.

Another more general comment, in the writing you write 'validation' while the heading reads 'evaluation'. Reading this section I am wondering if validation should not be replaced by evaluation in this section. L166: 'We preselect a set of stations' How did this pre-selection go, or is that what is described next? (if so then use something like 'to this end' instead of 'hereby'). (L167-170) I cannot follow this. L172: 'calibrated': there was nothing on a calibration before.

Thank you for this comment. We will change the term "validation" to "evaluation". Here, the pre-selection of stations refers to the stations with good agreement of observed and simulated ZWL percentage (percentage difference < 15%). We agree, that this is misleading as we show longitudinal connectivity for all stations in Figure 4 and later on, we use other criteria (direction change to zero leakage and ZWL occurrence) for the evaluation of the relationship of vertical and longitudinal connectivity. We will therefore change this passage.

The term calibration was used in L172 with respect to the determination of the zero water level threshold. In order to avoid misinterpretation we will change the phrasing.

[3] **Stresstest scenario definition** It is not completely clear what goes into the first stress 'changes in groundwater recharge'. Is this driven by climate input only or also by varying soil parameterization? I found the use of scenario somewhat misleading as 'scenario' refers to projections and potential future and the analysis is on the recent past (or current climate). So analysis or assessment might be a better term. Check throughout the manuscript how you refer to these tests. I also read the stress test (without scenario) this heading can then also read the stress test definition.

Thank you for this comment. We acknowledge that using both terms can be misleading. Thus, we will replace the term "scenario" in the manuscript and change the heading to "stress test definition". Regarding the distinction between stress tests and scenarios, we refer to Hellwig et al., 2021: "Unlike climate change scenarios which provide probabilities of changes based on specific projection assumptions, scenario-neutral stress tests explore the systems' general responsiveness, e.g. to other environmental changes or to extreme events. Therefore, stress tests must not be interpreted as predicting future conditions but rather as providing information on system responses for management or adaptation planning."

[4] **validation of zero water levels.** The first finding is that simulated groundwater levels (heads or depths?) are underestimated (too deep or too shallow?) but that this does not impact simulated discharge. The underestimation has to do with the parameterization of the river bed. It is not clear which part of the parameterization is impacting the results. Probably the drainage level (aka the boundary condition) is meant here as the river bed resistance (for example) does not impact groundwater heads that much. Results are not well explained here. More in general in the result section (here and other pasts) there are several parts that are redundant and describe methods instead of results, for example :L216, L223-226. L279-271.

Thanks for pointing out redundancies, which we will remove in a revised version. Also, we see that some of the sentences are too complex with unclear reference. Observed mean GW heads are represented in the model (L222). However the assessment of how good a performance this is, is not straightforward as the reference run is an assumed natural condition for which no observations exist, and the inclusion of the drinking water extractions while it is based on real data, those do not represent all groundwater extractions in the entire valley.

Water levels in the river are underestimated. We will check the phrasing in L213-226 in order to avoid misunderstandings and provide more clarity on any comparisons.

[5] L250: 'leakage' is the same as 'leakage flow' that was used before in this Alinea. And can you explain why simulated leakage flow is highly uncertain for areas with increasing flows? How did you evaluate the level of uncertainty? L252 "Which conditions they may change' only considering limited model choices of groundwater pumping and recharge right?

In L250, we write that simulated leakage flow is highly uncertain at the borders of the alluvial valley aquifer as there are abrupt changes in the hydrogeology and topography causing strong changes of the water table slope. This is visible in the appendix in Figure A3. In L250, "A3" should be in brackets "(Figure A3)". We are sorry for this mistake and will change this. We will also replace the term "conditions" here to clarify, that we refer to physiographic characteristics, such as slopes, topography and the location (upstream, downstream) itself.

[6] **Figure 4:** I do not fully understand this figure. What do the violins represent? Green is the reference (and not a green dot) and orange is the well scenario (and not an orange box) (see legend). What are the percentiles representing? What is the threshold? Where do 'losing' and 'gaining' come from? I can understand this but throughout the whole methodology and result this terminology was not used (but positive and negative leakage).

We regret this legend error and will change the legend.

The threshold is the location-specific threshold determined for zero leakage. This is mentioned in the following section 3.2.3 in L271-273. In the previous lines (L266-271) we

explain the terms 'losing' and 'gaining' conditions. For better understanding and readability, we will move this explanation to section 3.2.2

[7] L335-344: I disagree with the argument for not doing a sensitivity analysis. Also for a general interpretation of the result a sensitivity analysis, and varying parameter settings of a few key parameters, would have been useful to better understand what we learn from this modeling experiment (for example look at what is done in large-scale modeling studies). Also, I am a bit skeptical about the 'reduce calculation times' argument. How long does your model run the calculation times do not increase in when you re-run your model for different parameter settings, it is just more work. I would strongly recommend to rewrite your argumentation for not doing a sensitivity analysis or simply don't bother to explain.

We agree and also refer to our reply to R1. A model sensitivity analysis of varying parameter settings would indeed be useful to further investigate model uncertainties.

We agree, that the calculation time alone does not explain why we did not do a sensitivity analysis. An entire model sensitivity analysis would require changes of several parameters, which interact with each other. This effort would be worth a study of its own. The focus in this study however, was on stress tests responses towards different types of stresses. We will elaborate this further, as already suggested in our reply to R1.

Minor comments

[8] L105-107. In this sentence, it is not clear if 'baseflow' is the groundwater that is released to the stream or the constant flow in a stream that comes from the gradual release of groundwater. Also, what follows is confusing 'the degree of connectivity'; connectivity of what? Also, this suggests you discuss the groundwater discharge to the stream that contributes to rivers' baseflow. I recommend being as clear as possible on the terminology, and baseflow is a difficult term.

In the study by (Ott and Uhlenbrock, 2004) baseflow is the slow runoff component. Here, we mean the degree of connectivity between groundwater and surface water. This will be specified.

[9] **writing in general**. I recommend an English language check to correct grammar and common writing mistakes. For example, linking sentences together with 'and' where the start of a new sentence would be preferred; referring to previously mentioned aspects with 'this' or 'these' where it is not always clear where it refers to; misuse of commas and often no use of comma's where comma's are needed for the readability. Sometimes the meaning of the sentence also completely changes when a comma is not placed (I had the read some sentences several times to understand a comma was missing). The level of the manuscript will increase significantly if the writing is improved.

Thank you for this comment. We will revise the manuscript with respect to the English language.

L128: To represent both the surface and surface system**s**

Will be changed.

L123: well extractions à groundwater extractions

Will be changed.

L127: "The extension of the model domain"à the first part is confusing: the model domain covers.

Will be changed.

L129 '< comma> as well as'

Will be changed.

L131-133: 'The percolation … ': a clear example where commas should be placed or new sentence should start as now it reads as if you sum up recharge and the sum of interflow and overland runoff.

Will be changed.

L133: 'time invariant': constant or static is a more common way of writing.

Thankyou for the suggestion. We will consider to change the term.

L160-161: leave this out.

Will be changed.

L162: A direct validation is not possible

Will be changed.

L164: If I understand it correctly you compared observed water table heads and streamflow to simulated values. Thus, 'negative outliers of stream stages …. For gw heads far below the surface only concerns the simulated values? Not clear from the writing.

Yes, this concerns only the simulated values. The phrasing will be changed.

L245 is slightly *more* positive (*meaning* leakage in the natural system is lightly *higher)*

Will be changed.

L255-256: to stay consistent with the unit provided for the area. Also, a unit for leakage flow and specific leakage should be given.

We will add the units for specific leakage (mm/d) and leakage flow (m³/d) to the text. Units are provided in the respective Figures (Figure 3 and Figure 4).

L257 (setting the min…. ); I do not understand this. Is L271 similar (I can understand the latter).

L 257 explains how we normalized leakage. The brackets in L271 are thus a repetition and can be removed.

L274 modelled à simulated.

Will be changed.

L318: is it, not the other way around? Because of the resolution of your modflow model, you are not able to represent your drainage at a level of detail needed to accurately estimate the infiltration of losing streams. Which results in an underestimation of groundwater heads (or overestimation of depths).

Yes, exactly. The model resolution is too coarse, which is what is also explained here.

L321-327: this section hints at a hillslope effect but does not explain anything. (it turns in circles).

This section aims to express that uncertainties of simulated hillslope contributions could be a cause of the underestimation of zero water levels at upstream locations. We will change the phrasing.

L328: are weirs and bridges impacting the river bed to such an extent that it will impact your modeling more than, for example, the parameterization you use for riverbed conductance and the uncertainty related to that?

Weirs and bridges in the catchment area can significantly affect riverbed depth and slopes locally in a stream section. In addition, concrete basement of such structures can also impact riverbed conductance. Therefore, another potential source of uncertainty arises if such structures are neglected.

L332: not 'would be': a sensitivity analysis will be meaningful (and will be needed to properly interpret the modeling exercise.

Will be changed to "will be"

L427 "modeled leakage'

Will be changed.

L430: I have read this before, but here it says 'Due to the uncertainties we are not able to investigate the change in the magnitude'. Of course, you can study the changes in magnitude. You are not able to fully interpret the results, not because of the uncertainties but because of the lack of sensitivity analysis and/or observed or more reliable modeled data.

Will be changed to "we were not able to interpret the change in the magnitude"

---

## Author Response (AR1)

Response to Editor and Reviewers

Dear Editor,

we appreciate the opportunity to resubmit a revised version of our manuscript. The delay in our response, as already communicated, was due to the maternity leave of the lead author. In our revisions we addressed all comments by the two reviewers. In particular, we made the following improvements:
- general improvement of language and removal or better explanation of ambiguous terminology such as 'sensitivity analysis' (entire document)
- an improved and considerably extended description of the model details and the modelling approach in the method section under "Model Concept" and under "Reference simulation and stresstest scenarios" (formerly only 'stresstest scenario definition')
- revised research questions and aims and a revised focus and structure of the discussion in order to clarify the specific contribution and transferable knowledge developed in this study; in particular by highlighting the study's developed methodological approach of combining the modelling of stresses with specific metrics of stream intermittency for assessement of the responses.

In addition we improved the minor points of editorial nature that were raised and respond point-by-point below. Because the text edits are substantial, we were unable to create a track-change version of the manuscript. The dataset of streamflows from the already published experimental dataset of stream stages is available on request and currently prepared for publication as research data in case of acceptance of the paper. However, it is only used in the appendix of this manuscript and therefore not key data.

We are looking forward to your re-assessment.

Best regards
Amelie Herzog and co-authors

Response to Referee 1

Original comments in black and response in blue

Herzog et al. Study the longitudinal, vertical, and lateral connectivity of GW and SW in the Dreisam valley. The study is nicely done, generally well described and scientifically interesting. Mainly, I suggest adding some more explanation and discussion to better guide the reader and thus better help her to understand the storyline presented. I generally like the study and have no major comments beyond the constructive suggestions made below.

We thank Referee 1 for the positive feedback and the constructive comments below, which we considered in our revision as detailed below.

[1] There are a few editorial changes needed regarding the English. An example from the abstract: "This raises the question on" should be "question of". Or "By reason of the

physically-based" would better read as "Due to the phy…". Please have another read through the document for these instances.

The manuscript was revised and the English language edited (with a specific focus on prepositions etc.) and the abstract was considerably re-written to improve the storyline, remove redundance and to better explain how we use the case study to generate transferable methods and knowledge.

[2] In the abstract, "model reality" should probably be "model realism" to be more in line with the language used in other papers (e.g. Gharari et al. 2014 HESS; Hrachowitz et al. 2014 WRR; Wagener 2003 HP).

Revised.

[3] I like the abstract, but could you quantify terms a bit more. E.g. what is a short time scale in the context of this study? How many measurement locations did you consider? Etc. This is all discussed in the manuscript but might make the contributions of the study clearer right away if mentioned in the abstract.

The abstract was rewritten for conciseness and better storyline and now includes more information on the data.

The term "short timescales" mainly related to the model simulation time period (2014-2022). Therefore, it is not possible to assess long-term changes in GW dynamics due to GW withdrawals, which were already in place way before the year 2014. In the revised manuscript we replaced the term "short timescales" by the actual time period and rephrased the sentence.

Regarding the measurement locations, the model is evaluated with an available experimental dataset of zero water level measurements at 20 locations in the Dreisam valley, i.e. on several tributaries in the river network of the tributaries. However, for the evaluation of longitudinal and vertical connectivity, a prerequisite is that locations experience both, ZWL days and zero leakage in the model, which is the case at 7 locations. This is explained better in the methods and results now. We think all these details are a bit too much for the abstract. But the abstract was rephrased for more clarity regarding the use of observation data.

[4] It might be interesting to connect the metrics discussed, developed, and estimated in this study to the metrics (often called signatures) used in other studies. E.g. the recent study on reservoir impact in the UK by Salwey et al. (2023, WRR).

The employed approach to develop signatures and describe changes in these signatures due to water uses, (for example for water reservoirs in Salwey et al.) is indeed conceptually similar. In most studies, signatures/metrics are defined based on streamflow data at a point in the river. As we show however, the streamflow derived from the observed stages are not very accurate and there is only one official gauge at the outlet of the catchment. The objective in this study was to propose and use metrics that have support in our observations and which include information on vertical connectivity as well to better understand the streambed's drying dynamics. Therefore the metrics only consider dry phases (ZWL days) and connectivity changes (zero leakage, i.e. vertical connectivity) along the stream. The respective sections in the introduction and in the methods section were rewrittten for more

clarity  Future work or work in systems where models confidently simulate streamflow might consider signatures as in Salwey et al.. We added reference to the concept in the introduction.

[5] The authors state in lines 69ff: "While such parameter uncertainties are relevant when it comes to obtaining the best model results, they are less relevant if the focus is on process understanding." I do not agree with this statement. Understanding which parameter dominate system responses, and what preferred values they take when they do so, has long been part of assessing models regarding their physical realism (e.g. Reusser et al., 2009, HESS). So dismissing parameter uncertainty as a simple problem of model performance is really understating the problem. I therefore would expect a discussion of the potential influence of parameter uncertainty on the study outcomes in the conclusions or discussions sections. Even if a more detailed analysis is not feasible in this study.

We agree with this and acknowledge that the sentence was too simplified and misleading. We acknowledge that we used the term sensitivity not in the usual hydrological modelling terminology of a formalized (parameter) sensitivity analysis. We have changed and clarified this in the revised version and largely eliminated the term 'sensitivity' in favor of 'response to stress' (which is what we meant). See also responses to R2 -We have added a paragraph in the beginning of the discussion section on uncertainties and the prospects of carrying out a sensitivity analysis, which indeed is difficult (see below).

Late in the paper, the authors stated "The third research question addressed the sensitivities and changes of modelled connectivities in response to the applied stress test scenarios." Is this question really completely unrelated to parameter uncertainty? I can accept if the authors cannot add this element to this study, but a basic discussion of the potential influence would be good.

We agree that this research question would not be unrelated to parameter uncertainty (see previous comment) and changed the phrasing. Now, it should be more clear that in this analysis we focus on the potential of model stress-tests and did not perform a sensitivity analysis in the way this term is often used in modelling terminology, i.e. sensitivity to varying parameter values. The most important sources of parameter uncertainty that might affect model performance and also the stress test results are discussed in the beginning of the discussion section.

[6] I am afraid that I am a bit lost when looking at Figure 5. The super short caption is hardly helping me to understand what I am looking at here. The text discusses gaining and loosing conditions. Maybe making those explicit in the figure would be a start? More info please.

The caption text was adapted to express, that we are looking at examples of measurement locations which experience a direction change from gaining conditions (-1) to zero leakage (0).

[7] A general comment after looking at the next figure. Can you please make the captions more extensive. It is a bit annoying to have to go through much of the paper to look for abbreviations, variable names, location details etc. to understand figures. Please make the captions much more detailed so that the reader does not have to go through the text to understand the figure content. Or at least tell the reader exactly where to find the info needed to interpret the figure.

Captions (or where possible the legend) were revised and hopefully improved, also by more consistent terminology.

[8] Regarding the conclusions. I understand that the authors discuss what they specifically learn about their study region. However, it might be nice to add a short paragraph on what innovations, understanding or questions might be transferrable to other studies. What outcomes are general?

Thanks for pointing this out. In the revised version we worked out transferable knowledge gain more.

[9] As future work, the authors might want to consider a broader sensitivity analysis which could include both the stress test to the system as well as uncertainty in parameters or other model inputs. That would create a generic framework for analyses of the type presented here.

Thanks for this comment. Indeed, this would be the ideal follow up. We added this to the discussion and to the outlook on future work.

**Response to Referee 2**

Answers to the comments in blue.

I have read this discussion paper with great interest. I do not doubt the science that went into this study. However, the way it is presented is rather unsatisfying. This feeling covers all aspects of the paper, specifically how the methodology and results are presented as well as the interpretation of results and conclusion.

We thank Referee 2 for the feedback on our study. We acknowledge the need for improvement of the presentation of the methods section and parts of the results in particular. The methods section was rewritten and reorganized. It now includes first a more detailed description of the model and model setup, then a more systematic description of the model simulations performed and then the metrics. We also revised all other sections considerably and improved language and presentation throughout (also see Response to the Editor).

Overall I am a bit puzzled about how I should interpret this study. It starts off with a larger context. But, much more than a 'very general, first attempt to model SW-GW interactions in the Dreisam valley (L422-424)' it is not. I am questioning how much we learned 'to help gain insights into factors to consider in future modeling studies' (L424). What are these insights specifically and to what extent are the findings from this study transferable to other studies? (and is that not already done, for example, in the larger scale groundwater-surface water models available?).

We see that the phrasing of aims and insights and the discussion of transferability of the results were not clearly aligned and too generic. In the revised version we have better clarified how we see the role of the case study and what we aim to add in general.

I regret there is no proper sensitivity analysis done of at least the most important parameter settings that impact the groundwater-surface water interactions. However, I do understand

that this is probably not something the authors want to add to their analysis at this stage. I found all conclusions on uncertainties of the simulated leakage, because of the very limited model evaluation, somewhat hard to interpret.  I have several points of concern:

Thank you for this comment. As already noted in the response to R1, we acknowledge that we used the term sensitivity not in the usual hydrological modelling terminology of a formalized (parameter) sensitivity analysis. We have changed and clarified this in the revised version and respond to each of the specific points separately. The potential of a sensitivity analysis is there, but complicated due to some aspects that we now discuss in the beginning of the discussion section.

[1] **Model concept**. There are several aspects that I cannot fully understand. (L118) 'We used a combination of..'; I am not sure what this means. Are the models coupled in a way, or does one model use the outputs of the other, or did you use a combination of model results of the analysis? Not clear.

The Modflow model uses recharge and runoff data simulated with the RoGeR model as an input. Thus, we used an offline coupled model approach here. The method section was rewritten to include those details.

(L123-124) 'Surface water fluxes, well extractions, and recharge are added in the form of boundary conditions'. How does this work in Modflow6? As far as I know, you have to translate surface water fluxes to heads and use it as a general head boundary in Modflow. To be able to calculate heads from fluxes there needs to be some calculations using assumptions on river depth and width. What was used here? For wells in Modflow, you can define this as a specific pumping rate or specific head. Recharge is probably used as an upper boundary and as an input flux. Is capillary rise considered as well? Not enough information is provided to understand how the Modflow model is built and how boundary conditions are implemented.

All this information is now included in the revised section 'Model concept'.

In summary, in Modflow6, the surface water fluxes, well extractions and recharge are added by means of the SFR-package, the well package and the RCH packages. Mathematically, the so called stress packages are boundary conditions (see Modflow6 documentation Langevin et al., 2017 chapter 6). Capillary rise is not considered in Modflow but taken into account during the recharge calculation with RoGeR. In case of capillary rise recharge as the input flux for Modflow can also become negative. River depth and width are given in the streambed parameterisation. The data availability of these parameters is now mentioned. The well package used in this study uses a flow rate/pumping rate for each well (see Langevin et al., 2017 chapter 6). Data for each well was available from the regional water supplier.

(L129-136): 'the model' is this referring to the smaller Dreisam model or the bigger model? 'requires spatially distributed parameters determining' what kind of parameters? Determining surface and subsurface flows so I am assuming soil parameterization, elevation etc but this is not mentioned anywhere. (L134): as the aquifers are unconfined (at least I did not read otherwise) does it make sense to use specific storage (neglectable of unconfined conditions) compared to specific yields? (L156-157) 'based on Manning': I think this needs a bit more explanation (or reference to the original code and/or coupling between surface and groundwater).

Details are now included in the new Model concept section in the methods section.

(L142-158) A general remark for this whole section is that there is not a clear structure, especially not in this last part. It might help to show a small model conceptualization or flow chart: what goes in (parameterization, forcing) and what comes out (discharge, which is then used to calculate river heads), etc. I truly recommend going through this section again and making sure everything is very clear: which input data and where did you get it from, what is calculated in which model component, and how does it feed into the next model component? A proper understanding of the model setup is crucial to understanding and interpreting results as a reader (also, specifically as no sensitivity analysis is performed on any parameter settings. Then at least be clear and open on how you construct and parameterize your models).

We agree and improved the clarity of the entire methods section including a changed order or subsections, removal of redundant material etc.. (see previous responses).

Some additional questions: the RoGeR model was at daily resolution, how about the groundwater model and the river routing? Same for spatial resolution? And vertical discretization of both the hydrological and groundwater models is also not that clear to me. Human interactions are implemented by groundwater extraction only. And is there a return flow to surface water and/or groundwater I cannot follow that either.

The spatial resolution of the GW model is 100x100m. RoGeR's spatial resolution is much higher as it runs separately for homogeneous polygons regarding elevation and soil parameters. Recharge and runoff for all intersecting polygons are summed for each grid cell to generate Modflow input data. Both RoGeR and Modflow run on a daily resolution. Modflow's vertical discretization comprises four layers. Figure 1c) contains a cross-section of hydraulic conductivities and the vertical discretization along this cross-section. We made these points more clear in the revised manuscript and added the missing information.

In this study, GW extractions are the only human influences considered. We now explain this better in the revised version. We agree that there are other human influences, such as urbanisation, soil sealing, land use changes, or water withdrawals for irrigation, which can also influence recharge, groundwater heads and interaction of groundwater and surface water. However, in our study catchment of river Dreisam groundwater extractions for drinking water supply is by far the most important human influence and the only quantifyable one to date.

To assess other factors, model stress test approaches, such as presented in this study, can be adapted accordingly in the future. We added this as to the conclusion. For example, the Modflow6 drain package can also be used to implement agricultural drains and other stresses, which potentially modify the GW head.

Groundwater extractions are used for drinking water supply of the city of Freiburg, which is located downstream of the study area. Sewage plants for Freiburg are located downstream of our study area as well. Therefore, we can assume that there is no relevant proportion of return flow within our study area.

(L142-150) More generally, I have some trouble understanding the term 'leakage' in this study. As I understand it now, it is used here to describe groundwater drainage and river

infiltration. For me, leakage would mean more the unintended movement or loss of groundwater from its natural subsurface flow and not the dynamic interaction between groundwater and surface water that changes due to groundwater pumping. Maybe the terminology is something to verify with a groundwater expert as well.

Thank you for this comment. The objective of this study was to simulate GW-SW interaction. In this study, leakage therefore refers to the exchange flow between GW and SW in this study. We agree that the use of the term "leakage" is probably not ideal. However, we used the term here for consistency with the Modflow documentation we refer to therefore and prefer to keep it. We explain this better now.

[2] **Evaluation of model results**. Similar to the previous I have many questions. More in general I find it hard to understand what ZWL represents. A conceptual figure just showing a connected groundwater-surface water system and a disconnected system might already help.

We regret that the definition of ZWL is not clear. ZWL means, that the streambed is dry. This means, that an interruption of longitudinal connectivity takes place once ZWL occurs in a specific stream section. We explain this better in the revised version.

The direction of GW-SW-interaction depends on the relationship of vertical (leakage direction) and longitudinal connectivity (ZWL). We did not provide a conceptual figure here as we explained this in L182-186. The relationship of leakage direction and ZWL is additionally displayed in Figure 5. The legend was revised for more clarity.

Another more general comment, in the writing you write 'validation' while the heading reads 'evaluation'. Reading this section I am wondering if validation should not be replaced by evaluation in this section. L166: 'We preselect a set of stations' How did this pre-selection go, or is that what is described next? (if so then use something like 'to this end' instead of 'hereby'). (L167-170) I cannot follow this. L172: 'calibrated': there was nothing on a calibration before.

Thank you for this comment. The term "validation" was changed to "evaluation". Here, the pre-selection of stations refers to the stations with good agreement of observed and simulated ZWL percentage (percentage difference < 15%). We agree, that this is misleading as we show longitudinal connectivity for all stations in Figure 4 and later on, we use other criteria (direction change to zero leakage and ZWL occurrence) for the evaluation of the relationship of vertical and longitudinal connectivity.

The term calibration was used in L172 with respect to the determination of the zero water level threshold. This is now better explained.

[3] **Stresstest scenario definition** It is not completely clear what goes into the first stress 'changes in groundwater recharge'. Is this driven by climate input only or also by varying soil parameterization? I found the use of scenario somewhat misleading as 'scenario' refers to projections and potential future and the analysis is on the recent past (or current climate). So analysis or assessment might be a better term. Check throughout the manuscript how you refer to these tests. I also read the stress test (without scenario) this heading can then also read the stress test definition.

Thank you for this comment. We acknowledge that using both terms can be misleading. Thus, we replaced the term "scenario" in the manuscript and changed the heading to "stress test definition". Regarding the distinction between stress tests and scenarios, we refer to Hellwig et al., 2021: "Unlike climate change scenarios which provide probabilities of changes based on specific projection assumptions, scenario-neutral stress tests explore the systems' general responsiveness, e.g. to other environmental changes or to extreme events. Therefore, stress tests must not be interpreted as predicting future conditions but rather as providing information on system responses for management or adaptation planning."

[4] **validation of zero water levels.** The first finding is that simulated groundwater levels (heads or depths?) are underestimated (too deep or too shallow?) but that this does not impact simulated discharge. The underestimation has to do with the parameterization of the river bed. It is not clear which part of the parameterization is impacting the results. Probably the drainage level (aka the boundary condition) is meant here as the river bed resistance (for example) does not impact groundwater heads that much. Results are not well explained here. More in general in the result section (here and other pasts) there are several parts that are redundant and describe methods instead of results, for example :L216, L223-226. L279-271.

We have improved the text for more clarity .

[5] L250:  'leakage' is the same as 'leakage flow' that was used before in this Alinea. And can you explain why simulated leakage flow is highly uncertain for areas with increasing flows? How did you evaluate the level of uncertainty? L252 "Which conditions they may change' only considering limited model choices of groundwater pumping and recharge right?

In L250, we write that simulated leakage flow is highly uncertain at the borders of the alluvial valley aquifer as there are abrupt changes in the hydrogeology and topography causing strong changes of the water table slope. This is visible in the appendix in Figure A3. In L250, "A3" should be in brackets "(Figure A3)". We are sorry for this mistake and changed this . We replaced the term "conditions" here to clarify, that we refer to physiographic characteristics, such as slopes, topography and the location (upstream, downstream) itself .

[6] **Figure 4:** I do not fully understand this figure. What do the violins represent? Green is the reference (and not a green dot) and orange is the well scenario (and not an orange box) (see legend). What are the percentiles representing? What is the threshold? Where do 'losing' and 'gaining' come from? I can understand this but throughout the whole methodology and result this terminology was not used (but positive and negative leakage).

To make the figure more focused we removed the information regarding percentiles and adopted the legends and axes labels. Further, we ensured that all terms used in the figure are introduced in the method section. Also, the figure caption was clarified.

[7] L335-344: I disagree with the argument for not doing a sensitivity analysis. Also for a general interpretation of the result a sensitivity analysis, and varying parameter settings of a few key parameters, would have been useful to better understand what we learn from this modeling experiment (for example look at what is done in large-scale modeling studies). Also, I am a bit skeptical about the 'reduce calculation times' argument. How long does your model run the calculation times do not increase in when you re-run your model for different

parameter settings, it is just more work. I would strongly recommend to rewrite your argumentation for not doing a sensitivity analysis or simply don't bother to explain.

We agree and also refer to our reply to R1. A model sensitivity analysis of varying parameter settings would indeed be useful to further investigate model uncertainties.

We agree, that the calculation time alone does not explain why we did not do a sensitivity analysis. An entire model sensitivity analysis would require changes of several parameters, which interact with each other. This effort would be worth a study of its own. The focus in this study however, was on stress tests responses towards different types of stresses. We will elaborate this further, as already suggested in our reply to R1.

Minor comments

[8] L105-107. In this sentence, it is not clear if 'baseflow' is the groundwater that is released to the stream or the constant flow in a stream that comes from the gradual release of groundwater. Also, what follows is confusing 'the degree of connectivity'; connectivity of what? Also, this suggests you discuss the groundwater discharge to the stream that contributes to rivers' baseflow. I recommend being as clear as possible on the terminology, and baseflow is a difficult term.

In the study by (Ott and Uhlenbrock, 2004) baseflow is the slow runoff component. Here, we mean the degree of connectivity between groundwater and surface water. This was specified .

[9] **writing in general**. I recommend an English language check to correct grammar and common writing mistakes. For example, linking sentences together with 'and' where the start of a new sentence would be preferred; referring to previously mentioned aspects with 'this' or 'these' where it is not always clear where it refers to; misuse of commas and often no use of comma's where comma's are needed for the readability. Sometimes the meaning of the sentence also completely changes when a comma is not placed (I had the read some sentences several times to understand a comma was missing). The level of the manuscript will increase significantly if the writing is improved.

Thank you for this comment. The manuscript was revised with respect to the English language.

L128: To represent both the surface and surface system**s**

changed.

L123: well extractions à groundwater extractions

changed.

L127: "The extension of the model domain"à the first part is confusing: the model domain covers.

changed.

L129 '< comma> as well as'

L131-133: 'The percolation … ': a clear example where commas should be placed or new sentence should start as now it reads as if you sum up recharge and the sum of interflow and overland runoff.

changed.

L133: 'time invariant': constant or static is a more common way of writing.

Thankyou for the suggestion. We will consider to change the term.

L160-161: leave this out.

changed.

L162: A direct validation is not possible

changed.

L164: If I understand it correctly you compared observed water table heads and streamflow to simulated values. Thus, 'negative outliers of stream stages …. For gw heads far below the surface only concerns the simulated values? Not clear from the writing.

Yes, this concerns only the simulated values. The phrasing was changed.

L245 is slightly *more* positive (*meaning* leakage in the natural system is lightly *higher)*

*changed.*

L255-256: to stay consistent with the unit provided for the area. Also, a unit for leakage flow and specific leakage should be given.

We added the units for specific leakage (mm/d) and leakage flow (m³/d) to the text. Units are provided in the respective Figures (Figure 3 and Figure 4).

L257 (setting the min…. ); I do not understand this. Is L271 similar (I can understand the latter).

L 257 explains how we normalized leakage. The brackets in L271 are thus a repetition and were removed.

L274 modelled à simulated.

changed.

L318: is it, not the other way around? Because of the resolution of your modflow model, you are not able to represent your drainage at a level of detail needed to accurately estimate the infiltration of losing streams. Which results in an underestimation of groundwater heads (or overestimation of depths).

Yes, exactly. The model resolution is too coarse, which is what is also explained here.

L321-327: this section hints at a hillslope effect but does not explain anything. (it turns in circles).

This section aims to express that uncertainties of simulated hillslope contributions could be a cause of the underestimation of zero water levels at upstream locations. We will change the phrasing.

L328: are weirs and bridges impacting the river bed to such an extent that it will impact your modeling more than, for example, the parameterization you use for riverbed conductance and the uncertainty related to that?

Weirs and bridges in the catchment area can significantly affect riverbed depth and slopes locally in a stream section. In addition, concrete basement of such structures can also impact riverbed conductance. Therefore, another potential source of uncertainty arises if such structures are neglected.

L332: not 'would be': a sensitivity analysis will be meaningful (and will be needed to properly interpret the modeling exercise.

changed to "will be"

L427 "modeled leakage'

changed.

L430: I have read this before, but here it says 'Due to the uncertainties we are not able to investigate the change in the magnitude'. Of course, you can study the changes in magnitude. You are not able to fully interpret the results, not because of the uncertainties but because of the lack of sensitivity analysis and/or observed or more reliable modeled data.

changed to "we were not able to interpret the change in the magnitude"